# Synergistic activation by Glass and Pointed promotes neuronal identity in the *Drosophila* eye disc

Hongsu Wang [1], Komal Kumar Bollepogu Raja[2], Kelvin Yeung [2], Carolyn A. Morrison [1,6], Antonia Terrizzano [1,7], Alireza Khodadadi-Jamayran[3], Phoenix Chen[1,8], Ashley Jordan[1], Cornelia Fritsch [4], Simon G. Sprecher [4], Graeme Mardon [2,5] & Jessica E. Treisman [1] ✉

The integration of extrinsic signaling with cell-intrinsic transcription factors can direct progenitor cells to differentiate into distinct cell fates. In the developing *Drosophila* eye, differentiation of photoreceptors R1–R7 requires EGFR signaling mediated by the transcription factor Pointed, and our single-cell RNA-Seq analysis shows that the same photoreceptors require the eye-specific transcription factor Glass. We find that ectopic expression of Glass and activation of EGFR signaling synergistically induce neuronal gene expression in the wing disc in a Pointed-dependent manner. Targeted DamID reveals that Glass and Pointed share many binding sites in the genome of developing photoreceptors. Comparison with transcriptomic data shows that Pointed and Glass induce photoreceptor differentiation through intermediate transcription factors, including the redundant homologs Scratch and Scrape, as well as directly activating neuronal effector genes. Our data reveal synergistic activation of a multi-layered transcriptional network as the mechanism by which EGFR signaling induces neuronal identity in Glass-expressing cells.

How cell fates are specified is a central question in developmental biology. Although some fates are determined by direct lineage inheritance or stochastic choice, cell–cell signaling is the most common mechanism for assigning cells distinct fates. However, most signaling pathways are used in multiple developmental contexts and induce specific differentiation pathways through interactions with tissue-specific transcription factors[1]. These intrinsic factors may determine cellular identity, while extrinsic signals control the time of onset and spatial location of differentiation. For example, cells are committed to the eosinophil lineage by expression of GATA transcription factors, but eosinophil maturation and expansion require

interleukin-5 signaling[2]. Intrinsic and extrinsic inputs can be integrated at the transcriptional level; in spinal motor neuron differentiation, the retinoic acid receptor recruits the histone acetyltransferase CBP to activate chromatin on target genes of the intrinsic transcription factor Neurogenin 2[3]. Combinatorial regulation of differentiation by intrinsic factors and extrinsic signaling is thought to occur frequently during development, but there are few examples for which the mechanism of transcriptional integration has been studied at a genome-wide scale.

Differentiation of *Drosophila* photoreceptor neurons from eye disc epithelial progenitors is a well-characterized system in which external signals and intrinsic factors are known to play important roles.

[1]Department of Cell Biology, NYU Grossman School of Medicine, New York, NY, USA. [2]Department of Pathology and Immunology, Baylor College of Medicine, Houston, TX, USA. [3]Applied Bioinformatics Laboratories, NYU Grossman School of Medicine, New York, NY, USA. [4]Department of Biology, Université de Fribourg, Fribourg, Switzerland. [5]Department of Molecular and Human Genetics, Baylor College of Medicine, Houston, TX, USA. [6]Present address: 10x Genomics, Pleasanton, CA 94588, USA. [7]Present address: Biology of Centrosomes and Genetic Instability Team, Curie Institute, PSL Research University, CNRS, UMR144, 12 rue Lhomond, Paris 75005, France. [8]Present address: Department of Biology, Boston University, Boston, MA, USA. ✉e-mail: Jessica.Treisman@nyulangone.org

The first photoreceptor in each cluster to differentiate, R8, is specified by the proneural transcription factor Atonal (Ato)[4]. Differentiation of the other seven photoreceptors in an invariant sequence is induced by the epidermal growth factor (EGFR) ligand Spitz and other signals[5,6]. EGFR signaling leads to phosphorylation and activation of the ETS transcription factor isoforms Pointed-P2 (PntP2) and PntP3, which induce the expression of PntP1, a more stable isoform that is necessary for the differentiation of photoreceptors R1–R7[7–9]. However, EGFR signaling is also active in many other developmental contexts, where it turns on distinct sets of target genes[1]. The intrinsic zinc finger transcription factor Glass (Gl) is expressed in all eye disc cells beginning just prior to their differentiation[10], and is required for the expression of photoreceptor-specific genes[11–13] as well as genes expressed in non-neuronal retinal cells[14]. In *gl* mutants, fewer cells in each ommatidial cluster express the neuronal marker Elav, suggesting that Gl contributes to neuronal fate specification[12,13]. The requirement for both EGFR signaling and Gl to promote photoreceptor differentiation provides a good opportunity to determine how these extrinsic and intrinsic inputs are integrated. Previous studies have shown transcriptional regulation of several cell type-specific genes by both Gl and EGFR signaling, and an enhancer that drives expression of *prospero* (*pros*) in R7 photoreceptors is directly regulated by Gl and Pnt[15–17], suggesting that these transcription factors may interact in cell fate determination.

Here, we further characterize the *gl* loss of function phenotype by single-cell transcriptomic analysis and find that the later-born photoreceptor types R1, R6, and R7 are most severely affected. By misexpressing Gl in the wing disc together with activated Ras, a component of the EGFR signaling pathway, we show that the two factors synergize to induce ectopic neuronal differentiation. This effect is *pnt*-dependent, indicating that the synergy is at the transcriptional level and suggesting that Gl and Pnt activate a gene regulatory network (GRN) that contributes to photoreceptor differentiation. GRNs can assume a relatively "deep" hierarchically layered structure in which intermediate transcription factors relay the effects of the master regulators to the terminal structural genes, as in neural crest development[18], or a relatively "shallow" structure in which the master regulators directly control numerous terminal genes, as for γ Kenyon cells in the *Drosophila* mushroom body[19]. Using targeted DamID[20] to characterize the GRN architecture downstream of Gl and Pnt, we find that they directly regulate both intermediate transcription factors and downstream effector genes. Gl and Pnt share more than half of their genomic binding sites in developing photoreceptors, suggesting that these two transcription factors act synergistically on individual enhancers that control the expression of neuronal genes.

## Results

### gl differentially affects photoreceptor subtype differentiation

In mutants lacking the zinc finger transcription factor Gl, the number of neurons in each ommatidial cluster is reduced to a variable extent[12,13]. To understand which photoreceptor subtypes are affected by the loss of *gl* function, we performed single-cell RNA-Seq (scRNA-Seq) analysis of *gl*[60j] null mutant[10,11] white prepupal eye discs and compared the results to our previous characterization of wild-type eye discs[21]. Since the mutant and control datasets were generated at different times, we performed data integration in Seurat to enable a comparative analysis (Fig. 1a). A UMAP plot of the integrated data showed that all of the expected eye cell types were present in the *gl* mutant, but there was a severe reduction in the numbers of the late-born photoreceptors R1, R6, and R7 and of fully differentiated photoreceptors, and an increase in the proportion of undifferentiated cells assigned to the preproneural (Ppn), morphogenetic furrow (MF) and second mitotic wave (SMW) clusters (Fig. 1b). Trajectory inference on the photoreceptor clusters using Monocle 3 showed that in *gl* mutants,

the differentiation trajectories of R1, R6, and R7 were incomplete, missing part of the path that connects them to the differentiated photoreceptor cluster (Fig. 1c, d). The earlier-born photoreceptors showed a more subtle disruption of their trajectories at late differentiation stages (Fig. 1c, d). These data show that *gl* is important for the differentiation of late-born photoreceptors R1, R6, and R7, and for late photoreceptor differentiation in general.

We further analyzed the differentiation of photoreceptor subtypes in *gl* mutants, using known Gl target genes[12,13] as well as additional target genes identified by differential expression analyses comparing specific cell type clusters in *gl* mutant and wild-type eye discs (Supplementary Data 1). We also obtained consistent results by using principal component analysis (PCA) to compare the genes with the highest loadings along PC1 for each cluster (Supplementary Data 2), which are likely to be those that change their expression during the maturation of each subtype[21]. In R8, the early transcription factors Ato[4] and Senseless (Sens)[22] were still expressed in *gl* mutants (Fig. 1e, f, Supplementary Fig. 1b, c). However, the later R8-specific marker *CG42458*[21] was not expressed in the *gl* mutant (Fig. 1g, h), indicating that *gl* is required for the late differentiation of R8 cells. *gl* did not significantly affect the expression of *rough* (*ro*), which encodes a transcription factor active early in R2, R5, R3, and R4[23] (Fig. 1i, j), but was required for the expression of *CG7991* and *blanks*, later markers for this group of cells and for R1 and R6 (Fig. 1k–n). *Bar-H1*, which encodes a transcription factor that is expressed and required exclusively in R1 and R6[24], and the R7-specific gene *CAP*[21] were lost from these cells in *gl* mutants (Fig. 1o–r). *pros*, a previously reported Gl target expressed in R7 and cone cells[17], and other markers of differentiated cone cells, such as *Ninjurin C (NijC)*, showed strongly reduced expression in *gl* mutants (Supplementary Fig. 1a, f–i; Supplementary Data 1), consistent with the reduction in Cut-expressing cells observed in previous studies[13,14]. In addition, *gl* mutants showed greatly reduced expression of pan-photoreceptor markers activated late in differentiation, such as *CG34377* and *derailed* (*drl*) (Fig. 1s, t, Supplementary Fig. 1d, e). These results show that *gl* is required for the induction of some genes expressed in each of the photoreceptor types but has the greatest effect on the differentiation of R1, R6, and R7. It is possible that some of the defects in later photoreceptors are due to changes in the expression of genes that are required in early-born cells to induce their neighbors to differentiate, such as *rhomboid* and *roughoid*, which encode Spitz-processing enzymes, in R2 and R5 (Supplementary Data 1).

### Glass and EGFR signaling synergize to induce neuronal genes

Gl is expressed in all cells posterior to the morphogenetic furrow[10] and autonomously controls the differentiation of non-neuronal cell types as well as photoreceptors[14]. Its broad expression and function suggest that activation of the photoreceptor differentiation pathway must require additional input. EGFR signaling is known to be required for the differentiation of all photoreceptors other than R8[6], and R8 also initiates differentiation correctly in the absence of *gl* (Fig. 1e, f). To determine whether Gl and EGFR signaling interact, we ectopically expressed Gl and an activated form of the EGFR pathway component Ras[25] in the wing imaginal disc, which consists of epithelial progenitor cells similar to those in the eye disc but lacking retinal determination gene expression. While expressing Gl in clones in the wing disc can activate some photoreceptor-specific genes[14], it does not induce neuronal markers such as the RNA-binding protein Elav or the microtubule-associated protein Futsch[26,27] (Fig. 2a–c). Activating EGFR signaling by expressing hyperactive Ras[V12] induced overgrowth of the clones[25] but also failed to induce neuronal markers (Fig. 2d–f). However, expressing both Gl and Ras[V12] together induced strong expression of Elav and Futsch within the clones (Fig. 2g–i, m, n).

One possible mechanism for this synergy could be phosphorylation and activation of Gl by Mitogen-activated protein kinase (MAP

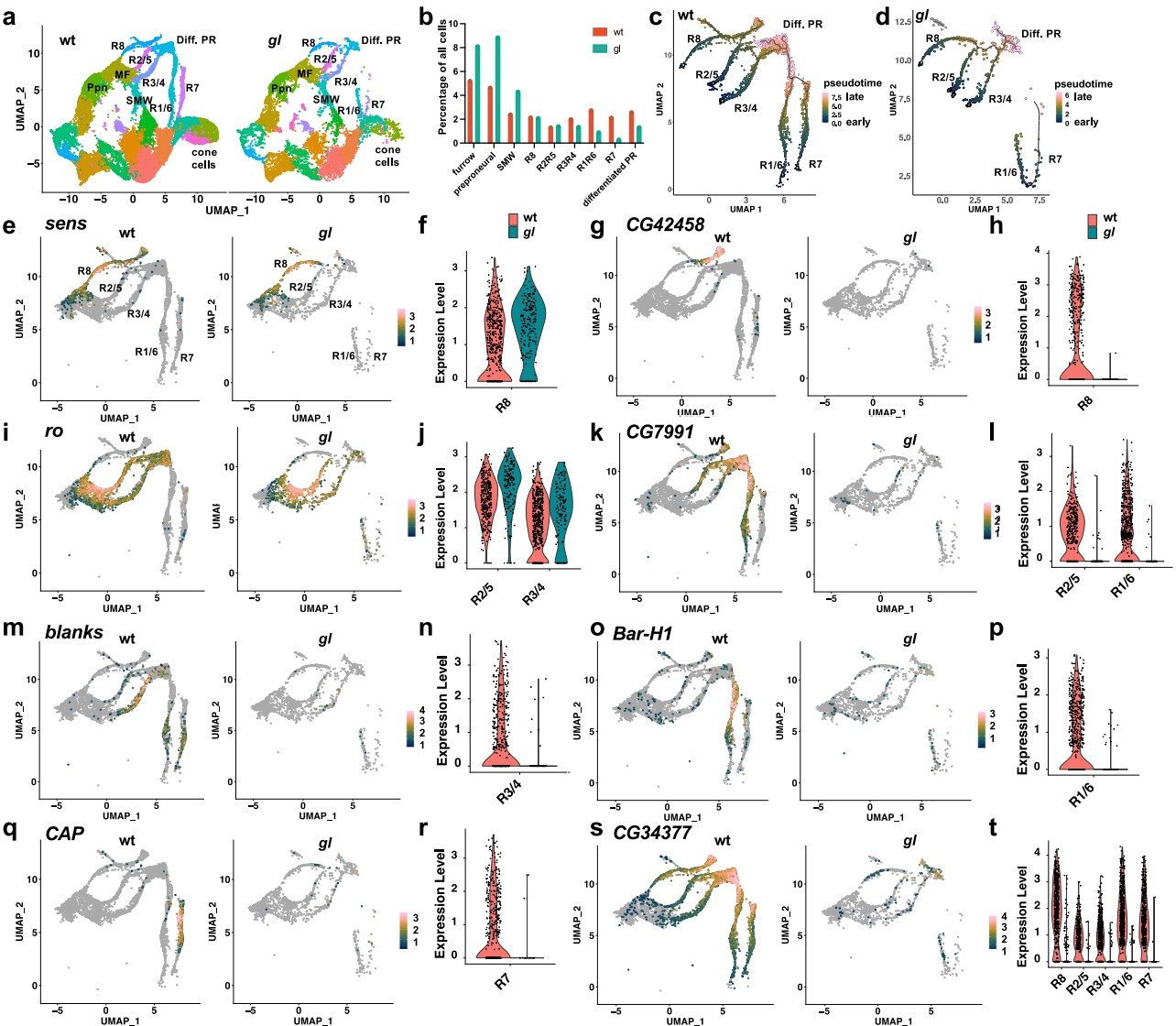

**Fig. 1 | scRNA-Seq characterization of photoreceptor differentiation defects in *gl* mutants. a** UMAP dimensional plot of scRNA-Seq data from wild-type and *gl*[60j] white prepupal eye discs harmonized using the Seurat integration method. The same colors are used for equivalent clusters in the two conditions. Clusters corresponding to photoreceptor types and cone cells are labeled. Other labeled clusters are Ppn preproneural, MF morphogenetic furrow, SMW second mitotic wave, Diff. PR differentiated photoreceptors. **b** Percentage of total cell counts in these integrated clusters in wild-type (red) and *gl* mutant (green) conditions. **c**, **d** Monocle3 trajectory inferences of the photoreceptor clusters using the integrated UMAP and raw counts for wild-type (**c**) and *gl* (**d**). Pseudotime is shown on a batlow scale, beginning with gray at the roots and ending with pink for the most differentiated cells. **e**, **g**, **i**, **k**, **m**, **o**, **q**, **s** Feature plots of gene expression levels in wild-type and *gl* mutant eye discs, with gray indicating no expression and pink the highest expression. **f**, **h**, **j**, **l**, **n**, **p**, **r**, **t** Violin plots of gene expression levels in the indicated selected cell clusters, with wild-type in red and *gl* in green. **e**, **f** *sens* is still expressed in R8 in *gl* mutants; **g**, **h** the late R8 marker *CG42458* is not expressed; **i**, **j** *rough* is still expressed in R2, R5, R3, and R4; **k**, **l** *CG7991* is lost from R1–R6; **m**, **n** *blanks* is lost from R3 and R4; **o**, **p** *Bar-H1* is lost from R1 and R6; **q**, **r** *CAP* is lost from R7; **s**, **t** the late marker *CG34377* is strongly reduced in all photoreceptors.

kinase), which acts downstream of Ras[7]. Gl contains one match to the canonical MAPK phosphorylation site PXSP[28] (Supplementary Fig. 2a). We mutated this site (PFSP 164–167) in the Gl protein into a non-phosphorylatable version (Gl[PFAP]) and a phospho-mimetic version (Gl[PFDP]) to compare to wild-type Gl[PFSP]. When expressed in the pouch region of third instar wing discs with *nubbin* (*nub*)-GAL4, all three proteins were expressed at similar levels as visualized with a V5 epitope tag, and all induced the Gl target genes *chaoptin* (*chp*) and *sallimus* (*sls*)[11,14] to the same extent (Supplementary Fig. 2b–g, j, m, p). If phosphorylation of Gl by MAPK were necessary for the synergy between Gl and Ras[V12], we would predict that the phospho-mimetic form would be sufficient to induce neuronal genes without Ras activation, and the non-phosphorylatable form would be unable to synergize with Ras[V12]. However, we found that like wild-type Gl[PFSP],

Gl[PFAP] and Gl[PFDP] clones in the wing disc failed to induce *elav*, *futsch* or *lozenge* (*lz*), a target gene of Gl and EGFR expressed in R1, R6, and R7[29–31] (Supplementary Fig. 2h–o, t-ab). Acidic amino acids do not always mimic the effect of phosphorylation. However, as Gl[PFAP] was able to synergize with Ras[V12] to induce ectopic expression of *lz, elav* and *futsch* (Supplementary Fig. 2q–s, ac-ae), the synergy cannot be explained by MAPK phosphorylation of Gl at this site.

An alternative possibility is that the synergy occurs at the transcriptional level. To test this, we co-expressed Gl and Ras[V12] in clones lacking all three isoforms of the EGFR transcriptional effector Pnt[7,9]. In *pnt*[Δ88] mutant clones, induction of the neuronal markers Elav and Futsch was greatly reduced (Fig. 2j–n). These results suggest that EGFR signaling synergizes with Gl at the level of transcription, through the ETS transcription factor Pnt.

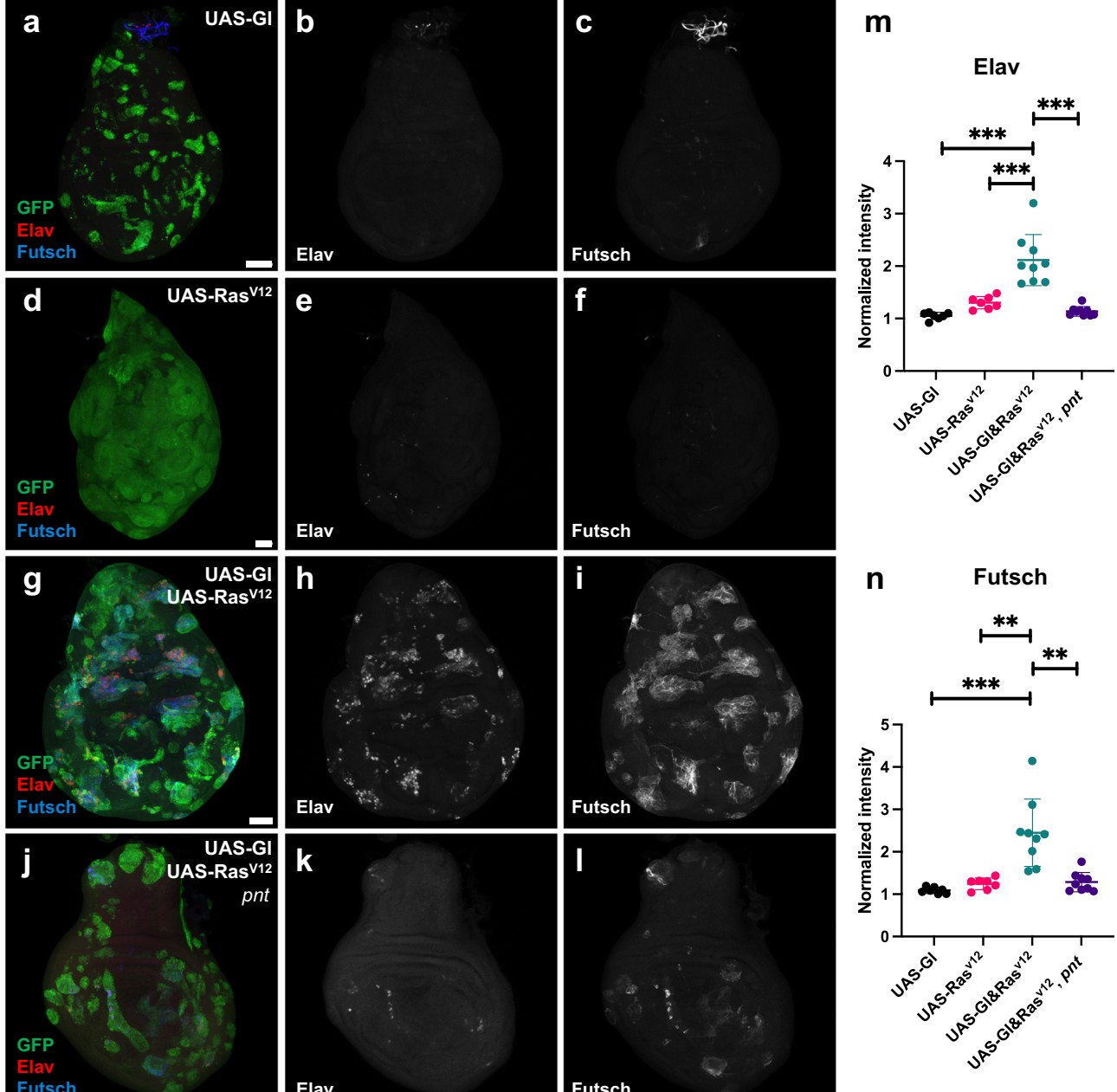

**Fig. 2 | Gl and Ras^V12 synergistically induce neuronal markers in the wing disc.**
**a–l** Wing discs in which clones overexpressing the indicated proteins are marked with GFP (green), stained for the neuronal markers Elav, a nuclear protein (**b, e, h, k**, red in **a, d, g, j**) and Futsch, a cytoplasmic protein (**c, f, i, l**, blue in **a, d, g, j**). Anterior is to the left and dorsal is up. **a–c** Gl overexpression does not induce either protein. **d–f** Ras^V12 induces clone overgrowth but not the neuronal markers. **g–i** Overexpression of both Gl and Ras^V12 strongly induces Elav and Futsch. **j–l** Overexpression of Gl and Ras^V12 in *pnt^Δ88* mutant clones induces much less Elav and Futsch expression than Gl and Ras^V12 in wild-type clones. All discs were imaged in parallel with the same laser settings. Scale bars, 50 µm. **m, n** Quantification of Elav (**m**) or Futsch (**n**) intensity in these genotypes, normalized to the background. Co-expression of Gl and Ras^V12 showed significantly greater Elav and Futsch levels than all other genotypes. Error bars indicate mean ± SD. **m** p(Gl + Ras, Gl) = 0.0002, p(Gl + Ras, Ras) = 0.0009, p(Gl+Ras, Gl + Ras *pnt*) = 0.0003. **n** p(Gl + Ras, Gl) = 0.0009, p(Gl + Ras, Ras) = 0.0018, p(Gl+Ras, Gl + Ras *pnt*) = 0.0021, two-tailed *t*-test with Welch's correction. *n* = 7 discs (*UAS-Gl*; *UAS-Ras^V12*); *n* = 9 discs (*UAS-Gl&Ras^V12*; *UAS-Gl&Ras^V12*, *pnt*). Source data are provided as a Source Data file.

To determine whether other neuronal genes could be induced by the combination of Gl and active Pnt, we performed RNA-Seq analysis on third instar larval wing discs with clones expressing *UAS-GFP* alone or in combination with *UAS-Gl*, *UAS-Ras^V12*, or *UAS-Gl* and *UAS-Ras^V12*, or with *pnt* mutant clones expressing *UAS-GFP*, *UAS-Gl*, and *UAS-Ras^V12*. As EGFR signaling in the wing disc is primarily transduced by the inactivation of the transcriptional repressor Capicua (Cic)[32], we also tested whether neuronal genes were activated when *UAS-GFP* and *UAS-Gl* were expressed in *cic* mutant clones. However, very few genes were more highly induced in this condition than in Gl-expressing clones alone (Supplementary Data 3), indicating that Gl-regulated genes rely on the activation of Pnt rather than the inactivation of Cic. Gene expression levels were normalized to GFP expression, which measures the proportion of the wing disc that is occupied by clones. Using differential gene expression analysis, 265 genes were found to be synergistically induced by Gl and Ras^V12 in a *pnt*-dependent manner with FDR < 0.1 and fold-change >2 (Fig. 3a, Supplementary Data 3). Only 34 genes

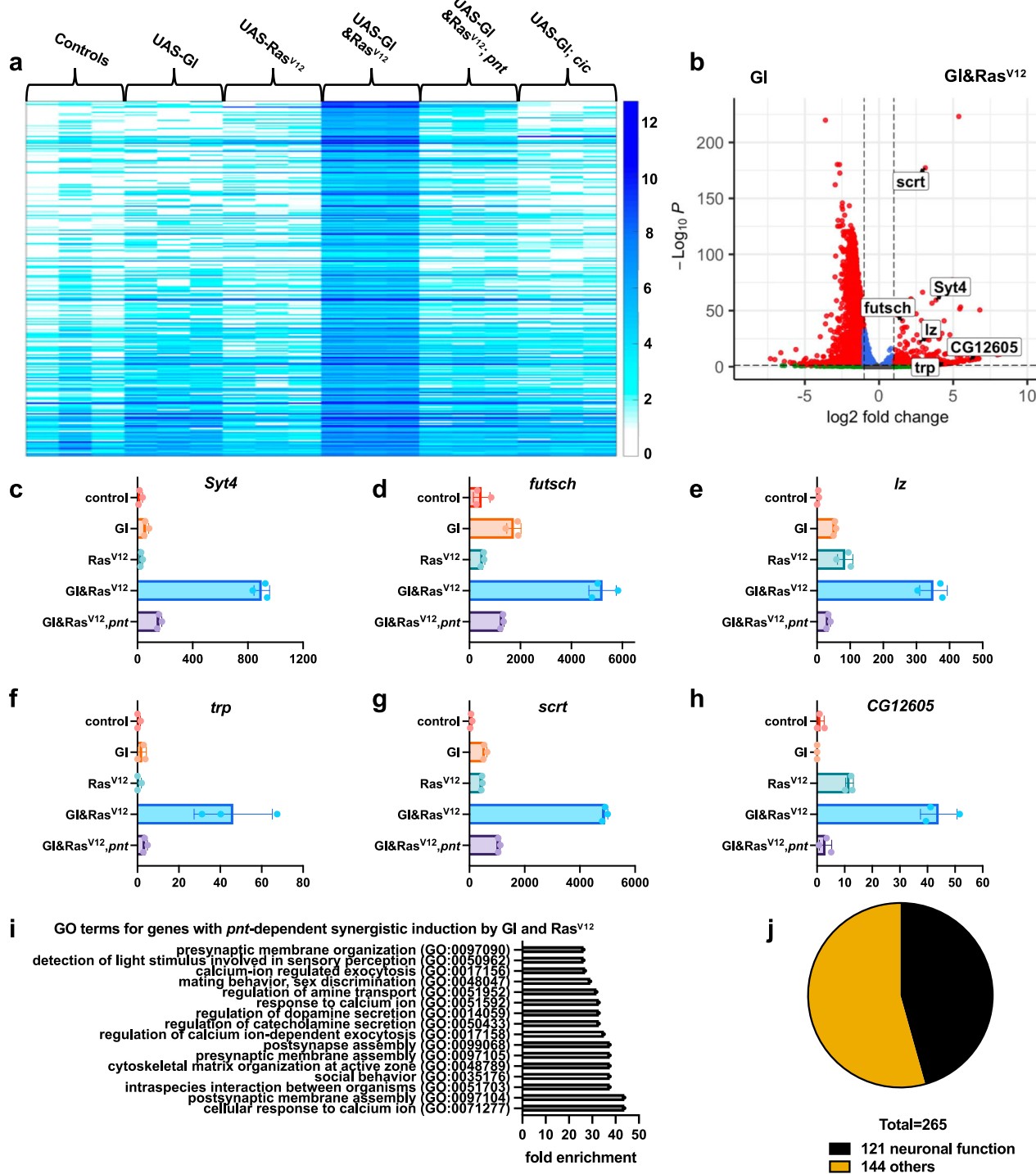

**Fig. 3 | Synergistic activation of a neuronal program by Gl and Ras$^{V12}$. a** Heat map showing the log$_2$-transformed expression levels of the 265 genes that were significantly upregulated in wing discs with clones expressing *UAS-Gl* and *UAS-Ras$^{V12}$* compared to *UAS-Gl* or *UAS-Ras$^{V12}$* alone by differential gene expression analysis and were attenuated in *pnt* mutant clones. **b** Volcano plot showing expression levels of genes that were significantly different with Gl and Ras$^{V12}$ co-expression compared to Gl expression alone. Genes with |log$_2$ fold change| > 1 are shown in red and <1 in blue. Selected neuronal genes are labeled. *p* values were computed with the Wald test in DESeq2. **c–h** Transcript levels (RPKM, libraries normalized to GFP level) of the synergistically activated neuronal genes *Syt4* (c), *futsch* (**d**), *lz* (**e**), *trp* (**f**), *scrt* (**g**), and *CG12605* (**h**). Error bars indicate mean ± SD. *n* = 3 biological replicates of the RNA-Seq experiment. **i** Panther-based GO term analysis of the 265 genes. Many GO terms are associated with neuronal functions. **j** Pie chart showing the proportion of the 265 genes that have neuronal functions documented in FlyBase. Source data are in Supplementary Data 3.

were synergistically induced independently of *pnt* (Supplementary Data 3). The genes that were induced by Gl and Ras$^{V12}$ but not by Gl alone included genes that encode phototransduction components such as *transient receptor potential* (*trp*), genes that encode

transcription factors such as *lz*, *scratch* (*scrt*)[33] and its homolog *CG12605*, and general neuronal genes such as *futsch* and *Synaptotagmin 4* (*Syt4*) (Fig. 3b). The expression of these genes was strongly increased in wing discs with Gl and Ras$^{V12}$ clones compared to either

factor alone, and the Ras[V12]-induced expression increase was greatly reduced in the absence of *pnt* (Fig. 3c–h). A gene ontology (GO)-term analysis of the 265 *pnt*-dependent synergistic genes showed many terms related to neuronal development and function (Fig. 3i), and 121 of the genes have neuronal functions or neuron-specific expression documented in FlyBase (https://flybase.org/) (Fig. 3j, Supplementary Data 3). These genes include some that are specific for subsets of photoreceptor types (Supplementary Data 3), and individual cells that express Gl and Ras[V12] appear to take on distinct identities (Supplementary Fig. 3a–c). These data suggest that Gl and Pnt synergistically induce a neuronal gene expression program in the wing disc. This synergy would not have been obvious from loss of function analysis, as *pnt* is essential for the recruitment of photoreceptors other than R8, making additional effects of loss of *gl* difficult to observe in *pnt* mutant clones (Supplementary Fig. 3d–l).

## Glass and Pnt bind to many common target genes

To examine whether Gl and Pnt can synergistically activate individual target genes, we used targeted DamID[20] to identify the sites bound by these transcription factors in the eye disc. We expressed Gl or PntP1 fused to *E. coli* Dam methylase, or Dam methylase alone as a control, either in eye disc cells at the onset of differentiation with *ato-GAL4*[34] or in cells specified as photoreceptors with *elav-GAL4*. Triplicate samples of each genotype were analyzed, and bound peaks were considered significant if the false discovery rate (FDR) was <0.1 and the fold change was >2 compared to the Dam-only control (Supplementary Data 4). *K*-means clustering (*k* = 8) was performed on the fold change of peaks that were significant in at least one genotype. In the resulting heatmap, more than half of the peaks were bound by both Gl and Pnt in *ato-GAL4*, *elav-GAL4*, or both conditions, while the remainder were specific to Gl or Pnt (Fig. 4a). A minimal enhancer of the *lz* gene that was previously shown to be a target of both Gl and the ETS factors Yan

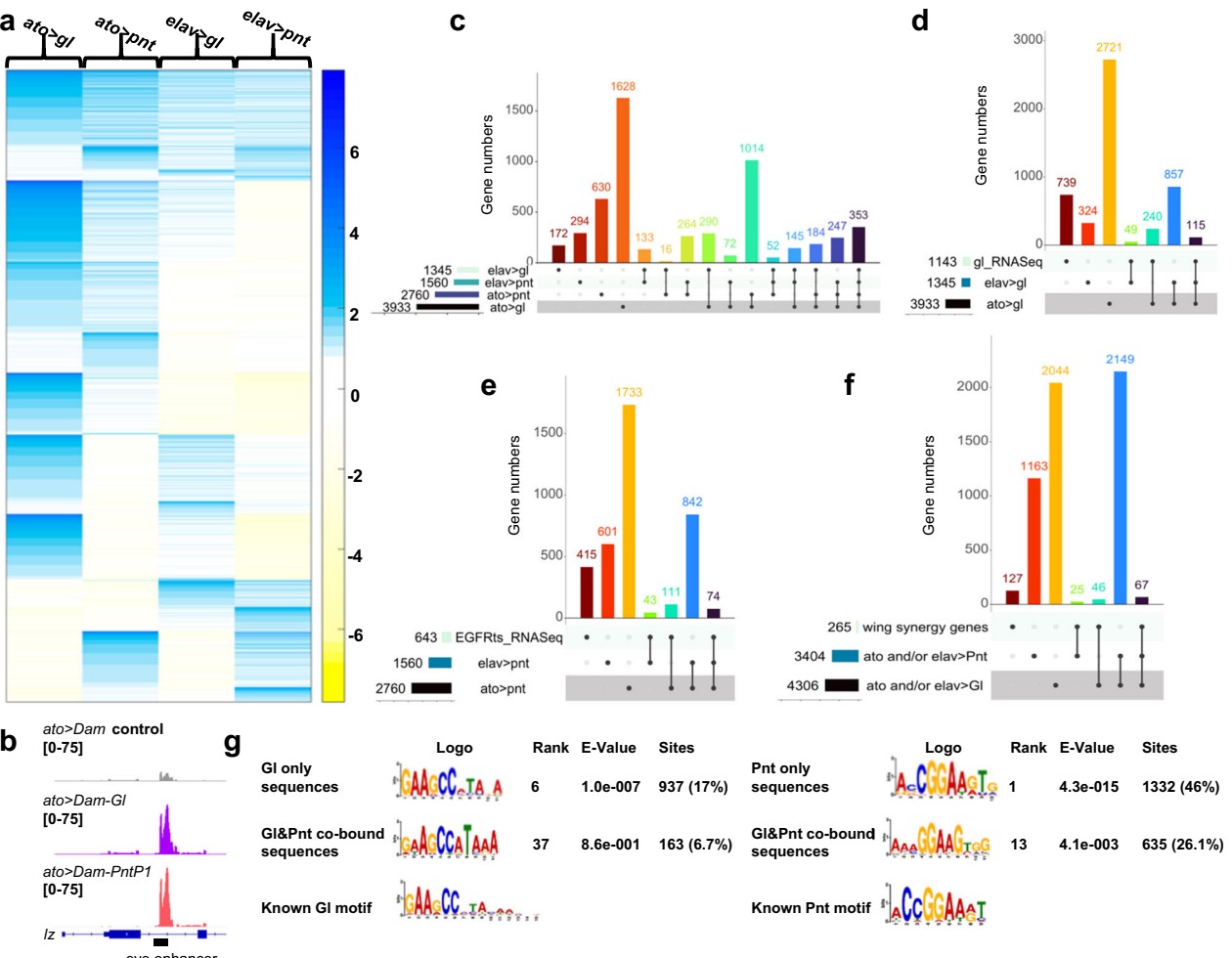

**Fig. 4 | Gl and Pnt bind shared target genes at two stages of photoreceptor differentiation. a** Log$_2$ fold-change compared to Dam control of peaks that were significantly bound by Dam-Gl or Dam-Pnt in at least one condition. The heatmap shows a *k*-means clustering, with *k* = 8. **b** Dam-ID peaks for *ato > Dam*, *ato > Dam-Gl*, and *ato > Dam-Pnt* on the second intron of *lz*, overlapping with the known Gl and EGFR-responsive minimal enhancer[30,31]. *n* = 3 biological replicates of each condition. **c** UpSet plot of genes that were significantly bound in at least one condition. Columns with dots connected by lines indicate binding in multiple conditions. The *y*-axis shows discrete gene numbers in each single set or intersection. **d** UpSet plot of genes significantly bound by Gl and their intersection with genes showing significantly reduced expression in *gl* mutant eye discs[35]. **e** UpSet plot of genes significantly bound by Pnt and their intersection with genes showing significantly

reduced expression in *Egfr[ts]* mutant eye discs. **f** UpSet plot of genes significantly bound by Gl or by Pnt in at least one condition and their intersection with genes synergistically activated by Gl and Ras[V12] in a *pnt*-dependent manner in the wing disc. Source data are in Supplementary Data 4. **g** Gl and Pnt motifs identified by STREME in peaks bound only by Gl, only by Pnt, or by both factors, showing the motif logo, the rank of that logo based on *E*-value compared to all logos found in the search, the *E*-value of the logo, and the number and percentage of peaks that contain the indicated logo. Both motifs appear at a lower rank and higher *E*-value in a smaller percentage of co-bound peaks compared to peaks bound only by that transcription factor. The Gl and Pnt binding motifs identified by Fly Factor Survey[39] are shown for comparison.

and Pnt[30,31] overlaps a peak that we found to be bound by both transcription factors in *ato-GAL4* cells, validating our DamID data (Fig. 4b). An UpSet plot of the intersections between sets of genes that were significantly bound in at least one condition (Fig. 4c) showed that the largest number of genes (3933) was bound by Gl in the *ato-GAL4* cells, and 51% of these were also bound by Pnt in at least one condition. Of the 2760 genes bound by Pnt in *ato-GAL4* cells, 68% were also bound by Gl in at least one condition. Similarly, 66% of the 1345 genes bound by Gl and 64% of the 1560 genes bound by Pnt in *elav-GAL4* cells were also bound by the other factor, and 353 genes were bound by both Gl and Pnt in both conditions (Fig. 4c). Gl and Pnt thus share many potential direct target genes.

To look for correlations between binding and functional regulation, we compared the genes that were bound by Gl to genes that showed altered expression in *gl* mutant eye discs in an RNA-Seq analysis[35]. Of the 1143 genes that were significantly down-regulated in *gl* mutants, 35% had Gl binding in at least one condition in our DamID dataset (Fig. 4d; Supplementary Data 4). Finding genes that are directly regulated by EGFR signaling is difficult because the failure of photoreceptor differentiation in mutants of EGFR pathway genes would have an indirect effect on the expression of many genes. To enrich for direct target genes, we used a temperature-sensitive allele of *Egfr*, *Egfr*[tsla 36], over a null allele. We compared *Egfr*[tsla]/*Egfr*[f2] third instar eye discs 24 h after a shift to the restrictive temperature of 29 °C to discs maintained at the permissive temperature of 18 °C (Supplementary Data 5). However, even known EGFR target genes like *argos* (*aos*)[37] showed only a small fold change under these conditions. Using a cutoff of $\log_2$ fold change > 0.25, $p < 0.05$, and standard deviation/mean < 0.5, to remove genes with high variability between samples, we found 643 genes that were down-regulated in *Egfr*[ts] mutant discs (Supplementary Data 5). Of these genes, 35% had Pnt binding in at least one condition (Fig. 4e, Supplementary Data 4). The lack of full overlap is not surprising, given the low sensitivity of the *Egfr*[ts] RNA-Seq experiment, the indirect effects of loss of *gl* or *Egfr* on downstream genes, and the potential binding of transcription factors to genes that are poised to be regulated later. Nevertheless, the DamID data showed direct binding of Gl and Pnt to a large proportion of the genes that they regulate. In addition, 67 of the 265 genes that were synergistically induced by Gl and Ras[V12] in the wing disc were bound by both Gl and Pnt (Fig. 4f).

To understand how Gl and Pnt interact with these genes, we used the MEME Suite program STREME[38] to discover enriched sequence motifs in the Gl and Pnt-bound peaks (Fig. 4g). A match to the canonical Gl motif identified by Fly Factor Survey[39] was found with high confidence in peaks bound only by Gl (*E*-value 1.0e−007), and a match to the canonical Pnt motif was also found with high confidence in peaks bound only by Pnt (*E*-value 4.3e−015) (Fig. 4g). For sequences that were co-bound by both Gl and Pnt, the motifs were found with much lower confidence (*E*-value 8.6e−001 for the Gl motif and 4.1e−003 for the Pnt motif), and the Pnt motif was also less similar to the canonical Pnt motif identified by Fly Factor Survey[39] (Fig. 4g). These differences suggest that cooperative binding might allow Gl and Pnt to recognize sequences distinct from their individually preferred motifs[40], providing a possible mechanism for the transcriptional synergy.

## Gl and Pnt binding correlates with chromatin accessibility

To examine the chromatin accessibility characteristics of Gl and Pnt co-bound sites, we first focused on the 265 genes that were synergistically activated by Gl and Ras[V12] in the wing disc in a *pnt*-dependent manner (Fig. 3a). In these genes, we found 58 peaks that were co-bound by Gl and Pnt in one or both conditions (Fig. 5c). We compared these peaks to wild-type single-cell ATAC-Seq data from the third instar eye disc[41]. Peaks bound by Gl and Pnt in *ato-GAL4*-expressing cells were most likely to be accessible to transposase tagging in progenitor cells in the morphogenetic furrow as well as in more differentiated

photoreceptors (Fig. 5a, d), while those bound by Gl and Pnt in *elav-GAL4*-expressing cells were most likely to be accessible in early or late photoreceptors (Fig. 5b, e). These results suggest that Gl and Pnt binding correlates with the stage at which the chromatin becomes accessible. We expanded this analysis to all Gl and Pnt binding sites using the pseudobulk ATAC-Seq signals corresponding to genome regions with significant binding (Supplementary Fig. 4a, b, Supplementary Data 4). Binding of Gl and Pnt in *ato-GAL4* cells correlated with high ATAC-Seq signals at early stages of differentiation, while binding in *elav-GAL4* cells correlated with high ATAC-Seq signals in late photoreceptors (Supplementary Fig. 4c–f). At both stages, the correlations were stronger for binding by Gl than by Pnt (Supplementary Fig. 4c–f).

Of all the peaks that were bound by both Gl and Pnt in at least one condition, 57.6% were in the upstream region, with the largest proportion within 1 kb of the transcription start site (Fig. 5f). Another 24.1% were in introns, with 8.8% in the first intron, a frequent site for regulatory elements in *Drosophila*[42]. Of the 2127 genes that had binding sites for both Gl and Pnt, Gl and Pnt bound to the same peak in 1752 (82%) of the genes (Fig. 5g). This high proportion of shared peaks would be consistent with cooperative binding of Gl and Pnt. This gene set is enriched for genes that are annotated with GO terms related to neuronal or eye development, as well as to the development of other tissues (Fig. 5h, Supplementary Data 4).

## Scrt and Scrape promote R7 differentiation downstream of Gl

Among the genes that were both bound by Gl and Pnt and synergistically induced by Gl and Ras[V12] in wing discs, 20 encode transcription factors that could act at an intermediate level of the neuronal induction GRN (Supplementary Fig. 5a). Some of these, like Lz, Runt, and Pph13, were already known to have functions in photoreceptor differentiation[29,43,44]. We used existing mutants or RNAi lines to test the remainder for defects in the adult eye, but most had no visible phenotype (Supplementary Fig. 5a), although aspects of photoreceptor differentiation that we did not examine could have been affected. One, *CG43347*, produced a small, rough adult eye when knocked down by RNAi with an eye-specific driver, *eyeless* (*ey*)*3.5-FLP, Actin > CD2 > GAL4* (Supplementary Fig. 5b). We generated a deletion allele that removed all the zinc fingers from this gene by CRISPR (Supplementary Fig. 5g). However, we found that homozygous mutants were viable with externally normal eyes (Supplementary Fig. 5c), and staining pupal or adult photoreceptors with Elav or Chp and cone cells with Cut revealed no defects (Supplementary Fig. 5d-f), indicating that the RNAi phenotype was probably due to off-target effects.

Another target of Gl and Pnt is *scrt*, which encodes a zinc finger transcription factor that is expressed posterior to the morphogenetic furrow in the eye disc[33]. Occasional missing photoreceptors were observed in *scrt* mutant adult eyes[33]. *scrt* has a paralog, *CG12605*, with a similar but weaker expression pattern in the third instar eye disc (Fig. 6a–d), which we have named *scrape* based on its homology and partial redundancy with *scratch*. Both *scrt* and *scrape* showed reduced expression in most photoreceptor types in *gl*[60j] mutant eye discs (Fig. 6a–d). DamID showed Gl and Pnt co-bound peaks in the intergenic region upstream of both *scrt* and *scrape* (Fig. 6e). *scrt and scrape* were also among the genes that were synergistically induced by Gl and Pnt in the wing disc (Fig. 3b, g, h). To investigate possible redundant functions of the two paralogs in the eye, we made a CRISPR deletion mutation in *scrape* on both a wild-type and a *scrt*[io11] chromosome (Fig. 6f). *scrape*[1E] single mutants were viable with no visible phenotype, and photoreceptor differentiation appeared normal when pupal retinas were stained for Elav and adult head sections for Chp (Fig. 6g–l). However, *scrape*[2A], *scrt*[io11] double mutant clones showed significant loss of R7 axons from the correct target layer in the medulla (Fig. 6r, u), a phenotype that was much less frequent in *scrt* single mutant clones (Fig. 6n, u). Staining for Pros and Elav in the 48 h APF retina showed normal R7 specification in 97% of double mutant ommatidia (Fig. 6s, t,

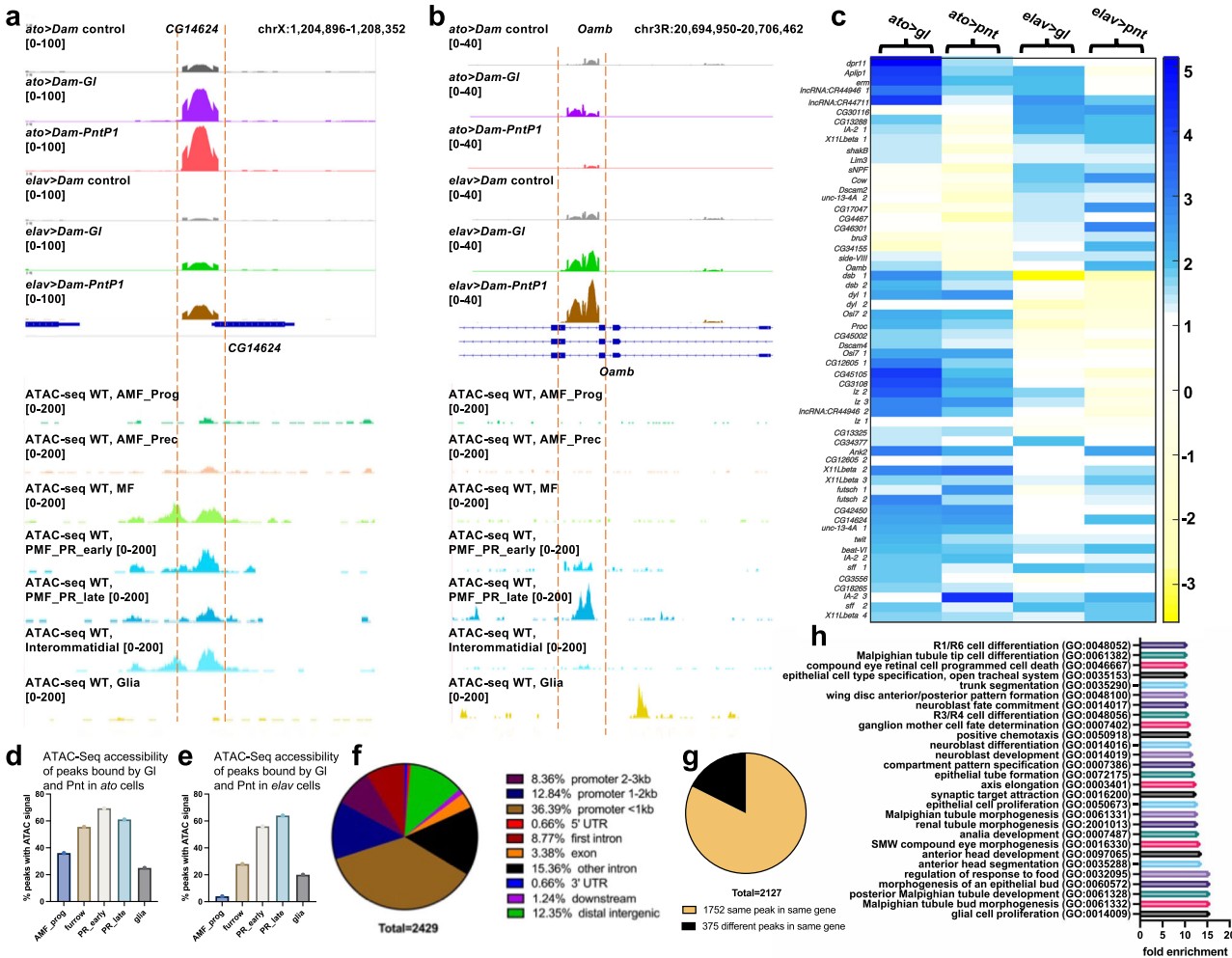

**Fig. 5 | Gl and Pnt co-binding correlates with photoreceptor-specific genome accessibility. a**, **b** Examples of a site bound by Pnt and Gl in *ato-GAL4* cells in the *CG14624* gene (**a**) and a site bound in *elav-GAL4* cells in the *Oamb* gene (**b**), aligned with ATAC-Seq data[41] for those genes in the eye disc. Peaks of interest are shown between the red dashed lines. Gl and Pnt binding in *elav-GAL4* cells correlates with accessibility in early (PMF_PR_early) and late (PMF_PR_late) photoreceptors, and binding in *ato-GAL4* cells also correlates with accessibility in the morphogenetic furrow (MF) and more weakly with anterior precursor cells (AMF_Prec). AMF_Prog, anterior progenitors. **c** A heat map showing k-means clustering (*k* = 5) of Dam-ID peak log$_2$ fold changes for peaks that were bound by both Gl and Pnt in *ato-GAL4*

and/or *elav-GAL4* cells in genes that were synergistically induced in the wing disc. **d**, **e** Percentages of the peaks in the heat map in (**c**) with shared binding in *ato-GAL4* cells (**d**) or *elav-GAL4* cells (**e**) that have the indicated ATAC-Seq accessibility profiles. Accessible peaks are defined as those with pseudobulk scATAC-Seq normalized counts ≥10. **f** Location distribution of 2429 peaks bound by Gl and Pnt in *ato-GAL4* and/or *elav-GAL4* cells relative to the nearest gene. **g** In 82% of genes that are bound by both Gl and Pnt, these factors bind to the same peaks. Source data are in Supplementary Data 4. **h** A Panther-based gene ontology search of the 1752 genes that contain these shared peaks showed enrichment of many terms related to neuronal or eye development.

v), suggesting that the loss of 56% of R7 terminals from the M6 layer (Fig. 6r, u) represents a defect in axon targeting rather than cell fate determination. These results rule out the possibility that Scrt and Scrape are sufficient to mediate all Gl and Pnt functions during photoreceptor differentiation, indicating that Gl and Pnt must act through a more extensive multi-layered gene regulatory network.

## Discussion

Cell differentiation requires differential gene expression, which is largely achieved by transcriptional regulatory networks. Gl and Pnt represent two different kinds of transcriptional inputs to eye disc progenitor cells. Gl is a cell-intrinsic transcription factor induced by the retinal determination genes Eyes absent and Sine Oculis[12] that carries organ identity information but is not cell type-specific[14]. Pnt is the effector of an extrinsic signaling pathway that provides a spatial and temporal cue to trigger the onset of differentiation[7,8]. We find that they act synergistically to induce the transcription of genes associated with neuronal differentiation. Although this interaction could, in

principle, be either direct or indirect, our finding that more than half of their binding sites in developing photoreceptors are shared strongly suggests a role for direct synergy. Interestingly, we extracted weaker Gl and Pnt binding motifs from these co-bound loci than from the peaks bound by each factor alone, suggesting that cooperative binding might help each factor bind to a sub-optimal motif, or that together they may recognize a distinct composite motif[40,45]. This would be consistent with the ability of the *Ciona* ETS homolog to bind to sub-optimal sequences when they overlap sites for the zinc finger transcription factor ZicL[46], and of mouse ETS1 to bind cooperatively with the zinc finger transcription factor GATA4 in endocardial cells[47]. Cooperative binding need not require direct protein–protein interaction; it can also be mediated by changes in DNA shape or nucleosome placement[48,49]. Alternatively or in addition, each factor may recruit different components of the general transcriptional machinery, leading to more robust activation of the co-bound genes[50].

At metamorphosis, the mRNA level of *pnt* decreases, terminating the synergistic effects of Gl and Pnt. Synergistic interaction between Gl

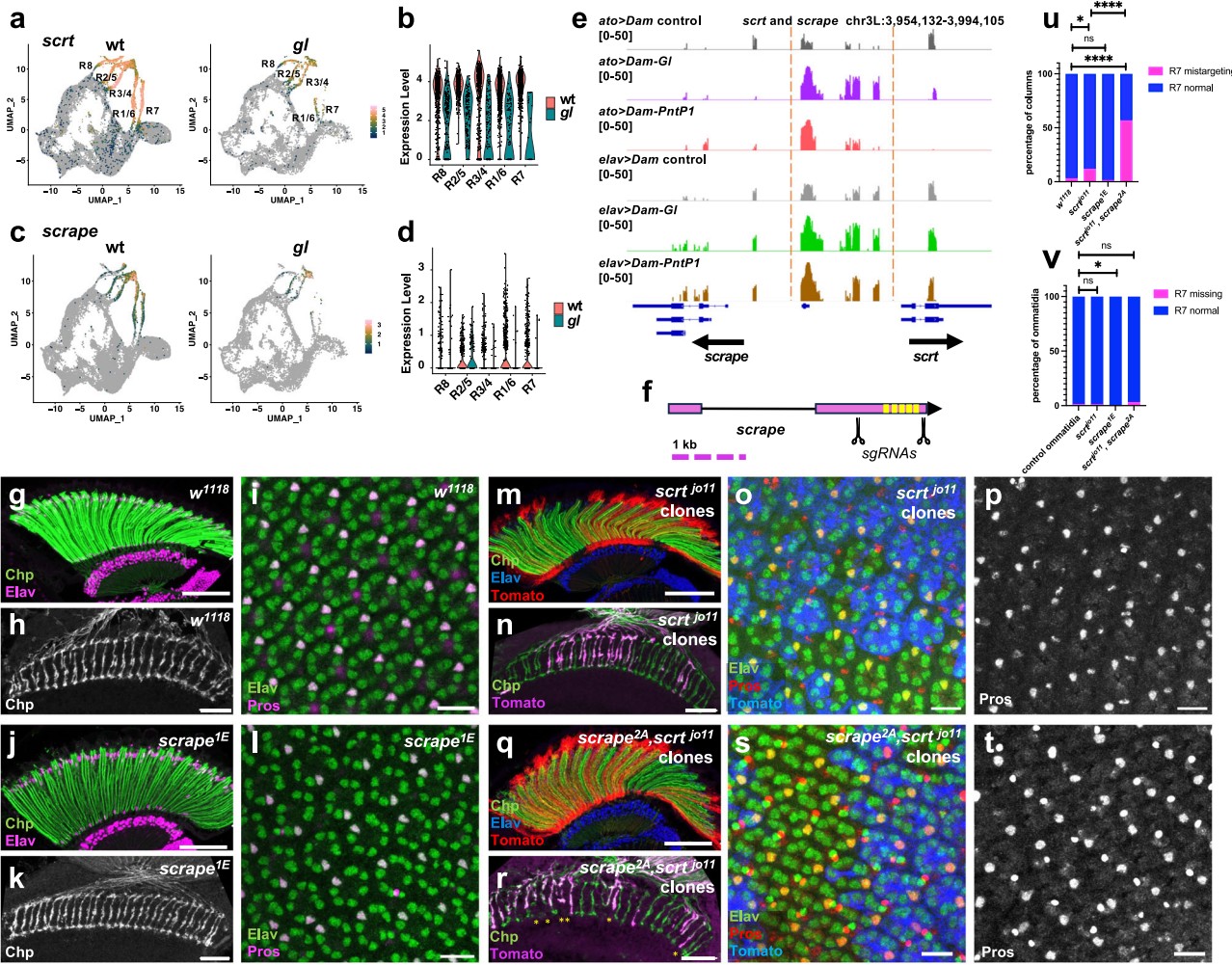

**Fig. 6 | Scrt and Scrape act redundantly in R7 photoreceptor axon targeting.**
**a, c** Feature plots and **b, d** violin plots showing that *scrt* (**a, b**) and *scrape* (**c, d**) are expressed in photoreceptors in wild-type eye discs and show reduced expression in *gl* mutant discs. **e** Dam-ID peaks showing Dam-Gl and Dam-Pnt binding to the intergenic region between the 5' ends of *scrape* and *scrt* (within the red dashed lines). **f** Schematic of sgRNAs used to delete all the predicted zinc fingers (yellow) of *scrape*. **g–i** *w^1118^*; **j–l** *scrape^1E^*; **m–p** *scrt^jo11^* clones; **q–t** *scrt^jo11^, scrape^2A^* clones. Clones are marked with myrTomato (red in **m, q**, magenta in **n, r**, blue in **o, s**). **g, j, m, q** Horizontal adult head sections were stained for Chp (green) to mark rhabdomeres and Elav (magenta in **g, j**, blue in **m, q**) to mark neuronal nuclei. **h, k, n, r** Horizontal sections of medullas stained for Chp (green in **n, r**) to mark R7 and R8 axons. Asterisks in (**r**) mark gaps in the R7 terminal layer corresponding to

*scrt^jO11^, scrape^2A^* clones. **i, l, o, p, s, t** 48 h APF retinas stained for Elav (green) and the R7 marker Pros (**p, t**, magenta in **i, l**, red in **o, s**). Most single and double mutant ommatidia contain an Elav and Pros-labeled R7 cell, and Pros is expressed equally strongly in wild-type and mutant R7s. Scale bars: 50 μm (**g, j, m, q**); 20 μm (**h, k, n, r**); 10 μm (**i, l, o, p, s, t**). **u** Quantification of the percentage of R7 axons that fail to reach the M6 layer in the medulla. n = 148 axon columns in 8 brains (148/8, *w^1118^*), 58/6 (*scrt^jo11^* clones), 144/5 (*scrape^1E^*), 65/7 (*scrt^jo11^, scrape^2A^* clones). ****p < 0.0001, *p = 0.0114, ns p = 0.34, Fisher's two-sided exact test. **v** Percentage of ommatidia that lack a Pros-labeled R7 cell. n = 292 ommatidia in 13 retinas (292/13, wt), 295/6 (*scrt^jo11^* clones), 606/9 (*scrape^1E^*), 208/7 (*scrt^jo11^, scrape^2A^* clones). p(wt, *scrt scrape*) = 0.2136; p(wt, *scrape*) = 0.042; p(wt, *scrt*) > 0.99, Fisher's exact test. Source data are provided as a Source Data file.

and Pnt in a limited time window could ensure that the network they activate is restricted to the phase of cell fate specification. Gl expression is maintained in order to carry out its later function of inducing a common set of terminal differentiation genes in the specified photoreceptors, in part through Pph13[12,13]. Gl also acts in non-neuronal cell types of the eye to promote their terminal differentiation, and our previous work suggests that it synergizes with distinct transcription factors in cone and pigment cells[14]. Interestingly, activated Ras did not appear to enhance the activation of most cone or pigment cell-specific genes in the wing disc beyond the level achieved by Gl alone (Supplementary Data 3), indicating that the synergy between Pnt and Gl may be limited to neuronal gene expression.

Transcription factors can induce differentiation by directly activating downstream effector genes, or by orchestrating a multi-layered transcriptional regulatory network in which intermediate transcription factors act on the effector genes. We found that intermediate

transcription factors act downstream of Gl and Pnt in photoreceptor differentiation; for instance, Lz, which promotes the differentiation of R1, R6, and R7, was already known to be a target of Gl and Pnt[30,31], and we showed that Scrt and Scrape are redundant effectors of Gl and Pnt that are required for normal R7 axon targeting. We identified additional likely intermediate factors based on significant Gl and Pnt binding detected by DamID, expression in photoreceptors, and altered expression in *gl* and *Egfr^ts^* mutants (Supplementary Fig. 6). These include the proneural factor Asense (Ase)[51]; Lim3, which is required for motor neuron subtype identity[52] and is expressed in R8; and Seven-up (Svp), which is needed for the differentiation of R1, R3, R4, and R6[53]. Figure 7 shows a model that includes some targets of these intermediate factors that have been identified either in the eye disc or in other tissues[54–58], as well as some effector genes that are directly bound and regulated by Gl and Pnt and known to contribute to neuronal functions such as synaptic specificity, synapse stability, and axonal

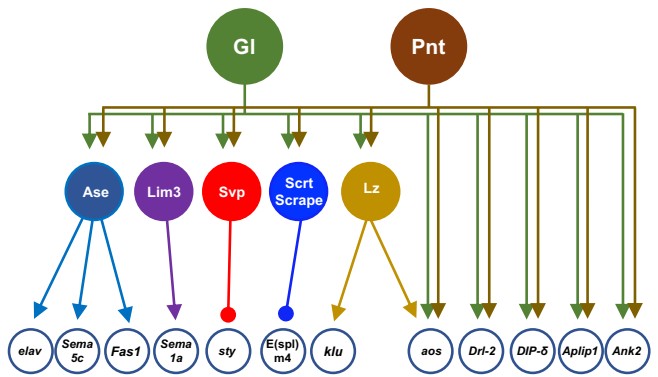

**Fig. 7 | Gl and Pnt act upstream of a multi-layered transcriptional network.** The diagram shows a subset of the transcription factors and effector genes that are directly bound by Gl and Pnt in our Dam-ID experiment and have known functions in photoreceptor development. Arrowheads indicate activation and circles repression. The effector genes shown to be targeted by intermediate transcription factors are based on previous studies[54–58]. Gl and Pnt directly activate some terminal effector genes that are relevant to eye/neuronal development and regulate others through an intermediate layer of transcription factors.

transport. These known and putative interactions suggest that Gl and Pnt activate an intermediate layer of transcription factors as well as directly regulating effector genes that contribute to neuronal differentiation.

Our results in the wing disc show that in an epithelial environment with no prior expression of retinal determination factors, Gl and activated Ras are sufficient to induce neuronal differentiation. They activate a large group of neuronal genes, including many that are bound by Gl and Pnt in the eye disc and which encode proteins such as ion channels, microtubule-binding proteins, and immunoglobulin family adhesion molecules. In most contexts, neuronal differentiation is induced by proneural transcription factors of the bHLH family. *ato* acts as a proneural gene that activates neuronal programs in R8[4], but no bHLH family member has been identified as a proneural gene for the other photoreceptors, although misexpression of Scute is sufficient to induce ectopic photoreceptors[59]. It is possible that *ase* has this role; it is the only member of the proneural gene family that we find to be expressed in nascent photoreceptors, although its expression in R3 and R4 is very low. It is directly bound by Gl and Pnt and induced by Gl and Ras[V12] in the wing disc, but it was excluded from our synergy list because Ras[V12] increased its expression less than twofold compared to Gl alone. If *ase* is not the proneural gene for R1–R7, it is possible that *gl* and *pnt* themselves take on the function of proneural genes in these photoreceptors.

The synergy between Gl and Pnt allows precise induction of neuronal differentiation by short-range cell–cell signaling within a field of cells poised to express eye-specific genes. Transcriptional synergy may increase the number of genes that can be activated in comparison to each factor alone, as well as elevate the expression level of key downstream genes. Such synergy could be a general mechanism by which to achieve precise and robust expression patterns. For example, human Matrix Metalloproteinase-1 (MMP-1) is synergistically activated by three transcription factors, c-Fos, c-Jun, and SAF-1, in a ternary protein complex[60]. These factors must activate MMP-1 only in response to inflammation, to avoid pathological outcomes such as cancer. The requirement for three transcription factors to activate MMP-1 demonstrates that transcriptional synergy can be a gatekeeper for sensitive processes such as extracellular matrix degradation. Moreover, the ternary complex can bind to enhancers with motifs for only one of the transcription factors, giving it the capability to robustly execute a markedly increased number of regulatory decisions[60]. It is possible that synergy between Gl and Pnt similarly increases their

regulatory power, as many of the co-bound peaks we detected had no recognizable motifs similar to the canonical Gl or Pnt-binding sites.

EGFR signaling shares many downstream intracellular components with fibroblast growth factor receptor (FGFR) and other receptor tyrosine kinases. FGFR signaling is also mediated by ETS-domain transcription factors related to Pnt, which are frequently found to engage in cooperative binding with other transcription factor families[48]. In the vertebrate retina, FGF signaling contributes to retinal progenitor proliferation, neuroprotection, and regeneration[61], and can regulate the initiation of neurogenesis by activating Atonal-related proneural transcription factors[62]. Our study provides a foundation to understand how ETS factors that mediate FGF signaling during vertebrate eye development might synergize with tissue-specific intrinsic transcription factors to transcriptionally orchestrate a precise and robust neurogenic program.

## Methods
### Drosophila genetics
*Drosophila melanogaster* was used for all experiments and analyzed at the developmental stages indicated. Males and females were used interchangeably except for the scRNA-Seq experiment, which used only males in order to sample genes on the Y chromosome. The stocks used for generating wing disc clones were: (1) *Ubx-FLP, UAS-GFP; tub-GAL4, FRT82, tub-GAL80/TM6B*[14] (2) *UAS-glRB; FRT82, UAS-Ras[V12], pnt[Δ88]/SM6-TM6B* (generated from FBal0346371, FBal0060587, FBal0035437, FBti0002074) (3) *UAS-glRB; FRT82, cic[Q219X]/SM6-TM6B* (generated using FBal0220444) (4) *FRT82, UAS-Ras[V12]* (5) *UAS-glRB; FRT82* (6) *UAS-glRB; FRT82, UAS-Ras[V12]/SM6-TM6B*. UAS-Dam-Gl was cloned by PCR amplification of the Gl-PA coding sequence from EST clone GH20219 (*Drosophila* Genomics Resource Center) with two primer pairs (CTGCGGCCGCACATGGGATTGTTATATAAGGGTTCCAAAC TC and CCCCGACTGCGAAAATCTGAGCAGGCAGAGCTTGCAC; GCTC TGCCTGCTCAGATTTTCGCAGTCGGGGAACTTG and GGCTCGAGTCA TGTGAGCAGGCTGTTGCC) to remove the unspliced intron of the RB isoform encoded by this cDNA. The two fragments were assembled in the pBluescript vector and then transferred into the Not I and Xho I sites of the pUAST-attB-LT3-Dam vector (GenBank KU728166). To make UAS-DamPntP1, *pntP1* cDNA[63] was amplified by PCR using the primers TAAGCACTCGAGATGCCGCCCTCTGCGTTTTTAG and TGCT TATCTAGACGCTGCTAATCCACATCTTTTTTCTC and cloned into pGEM-T-Easy, and a SpeI/XhoI fragment was cloned into the XbaI and XhoI sites of pUAST-attB-LT3-Dam. Injection for integration into the attP2 site and transgenic fly selection was done by Genetivision (Houston, TX). The fly lines used for the DamID experiment include: (1) *ato[3FL]-GAL4*[34], (2) *elav-GAL4/SM6-TM6B* (FBal0042579), (3) *UAS-Dam-Gl*, (4) *UAS-DamPntP1*, and (5) *UAS-Dam*[20]. For the *Egfr* RNA-Seq experiment, we analyzed eye discs from *Egfr[ts1a]/Egfr[f2]* (FBal0083481, FBal0003530) wandering third instar larvae that had either been maintained at 18 °C (control) or shifted to 29 °C 24 h earlier (*Egfr* mutant). To make the phosphorylation site mutants of Gl, the N-terminal V5 tag (GKPIPNPLLGLDST) and mutations to the MAPK consensus site were introduced by PCR using the *UASattB-gl-RA* plasmid as a template, and the primers TGAATAGGGAATTGGGAATTCC AACATGGGCAAGCCCATCCCCAACCCCCTGCTGGGCCTGGACTCCAC CATGGGATTGTTATATAAGGGTTCCAAACTC, GCGTGCTGGGCCGGG CATATGTCTT and CTGTTCCCATTCGACCCCTGCGG and CCGCAGGG GTCGAATGGGAACAG (PFDP), CTGTTCCCATTCGCCCCCTGCGG and CCGCAGGGGGCGAATGGGAACAG (PFAP) or CTGTTCCCATTCTCGCC CTGCGG and CCGCAGGGCGAGAATGGGAACAG (PFSP). This plasmid was then digested with EcoRI and NdeI and the amplicons were incorporated by Gibson assembly. Constructs were integrated into the VK37 attP site at position 22A3 by Genetivision.

RNAi lines knocking down possible intermediate transcription factors, listed in Supplementary Fig. 5a, were tested for adult eye phenotypes by crossing to *ey3.5-FLP, Act>CD2>GAL4; UAS-dcr2*

(generated from FBti0141243, FBtp0001640, FBal0211026). The *scrape* sgRNA sequences TGAGAACAGCCAGGACATTG and CCTGATGG GTGGCTCCTCGG, selected from www.flyrnai.org/crispr2/, were PCR amplified and cloned into pCFD5[64] via Gibson assembly. The construct was integrated into the attP40 site (FBti0114379) by Genetivision. Transgenic sgRNA flies were crossed to *nos-Cas9 ZH-2A* (FBti0159183) or to *nos-Cas9 ZH-2A; FRT80, scrt[iO11]/TM6B* (FBal0046413) and the F2 generation were screened by PCR. Three lines with the expected 887 bp deletion removing all the zinc fingers were recovered from each cross. The MARCM clone stock used to make *scrt[iO11]* single or *scrape[2A], scrt[iO11]* double mutant clones was *ey-FLP; gl-lacZ; lGMR-GAL4, UAS-myrTomato/CyO; FRT80, tub-GAL80/TM6B* (generated using FBti0015982, FBti0015985, FBti0058798, FBti0131969, FBti0002073, FBti0012693).The *scrape[1E]* single mutant was homozygous viable and was compared to *w[1118]* (FBal0018186) controls. The *CG43347* sgRNA sequences GCGAACACGCCGGTCACATT and CGTTCATGGCGGCCGC ATGG, selected from www.flyrnai.org/crispr2/, were cloned into pCFD5, integrated into the attP2 site (FBti0040535) by Genetivision and crossed to *nos-Cas9/CyO* (FBti0199256) flies. PCR screening of the F2 generation identified four lines with the expected 3.95 kb deletion in *CG43347*.

## 10x Genomics single-cell RNA-Seq

30 male white prepupal *gl[60j]* [11] eye discs were dissected, pooled, and dissociated into single cells as described[21]. Briefly, eye discs were dissected into ice-cold Rinaldini solution with 1.9 μM Actinomycin-D. Single cells were dissociated from eye discs mechanically by pipetting and enzymatically with Collagenase (100 mg/ml; Sigma-Aldrich #C9697) and Dispase (1 mg/ml; Sigma-Aldrich #D4818). Dissociated single cells were filtered and washed once with 0.05% Bovine Serum Albumin (BSA) in the Rinaldini solution. Cells were resuspended in 0.05% BSA in Rinaldini solution to a concentration of 1000–1200 cells/μl. Only samples with over 95% viability (assayed with Hoechst propidium iodide) were used for scRNA-seq. The 10x Genomics Chromium Next GEM Single-Cell 3′ Reagent Kit 3v31 was used to generate single-cell libraries. cDNAs generated were sequenced with NovaSeq 6000 (Illumina) to a depth of 477 million reads. Dissociation and scRNA-Seq of cells from wild-type white prepupal eye discs using the same protocol have been previously described[21].

## Bioinformatics analysis of scRNA-Seq data

FastQ files generated from sequencing were initially analyzed using the Cell Ranger v6.0.1 count pipeline and the *Drosophila melanogaster* reference genome Release 6 (Dm6). The median number of genes per cell was 2093, with 44,885 mean reads per cell. The filtered gene expression matrices from Cell Ranger were used as input in Seurat v4.2.1 to perform quality control and other downstream analyses. Potential multiplet cells and cells that showed high mitochondrial content (>40%) were removed. Further, only cells that showed a total number of genes between 200 and 5000 were retained. The filtered cells were normalized and scaled using Seurat SCTransform algorithm. The data were then reduced to the top 50 dimensions using principal component analysis (PCA) and Uniform manifold approximation and projection (UMAP). The data were clustered using FindNeighbors and FindClusters functions in Seurat to generate a cluster plot. Small clusters that showed artifactual gene expression (multiplets) and cells from the antennal disc (identified by expression of *Distal-less*), glia (*reversed polarity*) and brain (*found in neurons*) were removed, and 10,225 cells from the *gl[60j]* eye disc were retained.

Next, these cells were merged into a combined dataset with 26,669 cells from wild-type white prepupal eye discs[21], with condition IDs added. Seurat data integration was performed on the merged dataset[65]. Briefly, data normalization and finding variable features were run for both conditions. Integration features were selected, integration anchors were found, and data were scaled. PCA was performed, and

the first 50 dimensions were used for the FindNeighbors function. The resolution for the FindClusters function was set at 0.55, which was sufficient to separate different photoreceptor clusters. UMAP was run on the first 50 dimensions using seed 12 for the ideal orientation of the graph. After data integration, comparable clusters were shown in corresponding colors after dimensional plotting of the two conditions.

Each photoreceptor subtype was subclustered and PCA was performed using the Seurat RunPCA function. The eigenvalues and loading weights were then calculated for PC1 of each photoreceptor subtype. Significant genes in PC1 were retained by filtering genes with loading weights >0.01 or <−0.01. We compared genes with top loading weights in the first principal component (PC1) between wild-type and *gl* mutant and found that 50% of the top PC1 genes were common between them, while the remainder were found in one condition but not the other (Supplementary Data 2). Since the differentiation of R7 and R1/6 is severely affected in *gl* mutants, these subtype clusters were not well resolved and showed identical genes in PC1.

The package "scCustomize" was used to plot gene expression levels in the integrated UMAP using raw RNA counts (https://zenodo.org/records/10161832). Function FeaturePlot_scCustom provided the same scaling of gene expression level for both wild-type and *gl* mutant conditions. The "batlow" colormap (https://www.fabiocrameri.ch/batlow/) was used in the gene expression plots. Integrated data were split into wt and *gl* mutant using "SplitObject" prior to Monocle 3 trajectory inference[66]. "DefaultAssay" was set to "RNA" instead of "integrated" to allow trajectory analysis to be done using unintegrated raw RNA counts. The package "SeuratWrappers" was used to convert Seurat data containing integrated UMAP graphic information to a format readable by Monocle 3. Photoreceptor cells were chosen with the Monocle 3 function "choose_cells". Cells were re-clustered using raw data using the method "Louvain", using $K = 75$ for the wild-type to distinguish all PR types from each other and $K = 10$ for *gl* mutant (a lower value because the raw RNA data did not distinguish well between R1, R6, and R7). Default "learn_graph_control" parameters were used in learn graph function, and "use_partition" was set to "TRUE". Root nodes were manually added with the "order_cells" function. The same custom colormap as above was used for the pseudotime color scale.

## Immunostaining

Wandering third instar larval wing discs or 48 h APF pupal retinas were dissected in PBS and fixed in ice-cold 4% formaldehyde in PBS for 30 min (or 15 min for anti-Pros staining). After a 15 min wash with PBS/0.3% Triton X-100 (PBST), and for anti-Pros, a 1 h block with 10% donkey serum in PBST, samples were incubated overnight at 4 °C in primary antibodies in 10% donkey serum in PBST. After three 20 min washes with PBST, samples were stained for 2 h at room temperature in secondary antibodies in 10% donkey serum in PBST. Samples were washed again 3 times for 20 min with PBST and mounted in 80% glycerol in PBS. For cryo-sectioning, adult heads were fixed for 3–4 h at 4 °C in 4% formaldehyde in 0.2 M sodium phosphate buffer pH 7.2 (PB) after removing the proboscis. Heads were glued onto glass rods using nail polish, transferred through 20 min each in 5%, 10%, 25%, and 30% sucrose in PBS, and frozen in OCT compound. 12 μm cryosections were cut at −21 °C and slides were fixed post-sectioning for 30 min in 0.5% formaldehyde in PB at room temperature. After three 10 min washes in PBS/0.2% Triton X-100 (PBT), slides were blocked for one hour and incubated in primary antibodies overnight at 4 °C in 1% BSA in PBT. After three 20 min washes in PBT, slides were incubated in secondary antibodies in 1% BSA in PBT for 2 h at 4 °C, washed again in PBT, and mounted in Fluoromount (Southern Biotech #0100-01) for better preservation of the fluorophore[67]. Primary antibodies used were chicken anti-GFP (1:200, Invitrogen A10262), rat anti-Elav (1:50; Developmental Studies Hybridoma Bank (DSHB) Rat-Elav-7E8A10), mouse anti-Futsch (1:20, DSHB 22C10), mouse anti-Pros (1:10, DSHB Prospero MR1A), rabbit anti-dsRed (1:400, TaKaRa Living Colors®

Polyclonal 632496), and mouse anti-Chp (1:25; DSHB 24B10). Secondary antibodies were Jackson ImmunoResearch Cy3 and Cy5 conjugates used at 1:100 and Invitrogen Alexa488 conjugates used at 1:200. Samples were imaged with a Leica SP8 confocal microscope using a ×20 air objective lens for wing discs and a ×63 oil immersion lens for retinas and cryosections. Fluorescent intensities were quantified in ImageJ as intensity in clones divided by background intensity in the same wing disc, using a macro to enable multi-selection of every clone ROI in a given wing disc.

## RNA-Seq of wing and eye discs

Wandering third instar larval wing discs or eye discs were dissected from 20 animals per replicate with three replicates per genotype. RNA was extracted using Trizol (Invitrogen) followed by genomic DNA elimination and further purification with a Qiagen RNeasy Plus Micro kit. RNA quality and quantity were assessed using a Bio-analyzer 2100 (Agilent) prior to library preparation. Library preparation, sequencing, and quality checks were performed as previously described[67]. Sequencing reads were mapped to the reference genome using the STAR aligner (v2.5.0c). Alignments were guided by a gene transfer format (GTF) file. The mean read insert sizes and their standard deviations were calculated using Picard tools (v.1.126) (http://broadinstitute.github.io/picard). The read count tables were generated using HTSeq (v0.6.0), normalized based on their library size factors or by GFP expression levels using DEseq2, and differential expression analysis was performed. The read per million (RPM) normalized BigWig files were generated using BEDTools (v2.17.0) and bedGraphToBigWig tool (v4). To compare the level of similarity among the samples and their replicates, we used two methods: principal-component analysis and Euclidean distance-based sample clustering. All the downstream statistical analyses and generating plots were performed in the R environment (v3.1.1) (https://www.r-project.org/). Genes were considered significantly changed and were included in the heatmap if the |log2 fold change| was greater than 1 and FDR < 0.1 for wing disc RNA-seq. Genes were considered significantly changed and were included in the discussion model if the $\log_2$ fold change was >0.25, $p < 0.05$, and standard deviation/mean>0.5 for $Egfr^{ts}$ mutant larval eye discs. The heatmap was constructed using Matlab (R2022b). Wing disc samples were normalized using GFP levels except for $Ras^{V12}$ discs; these were normalized by library size because GFP normalization seemed to excessively downscale the gene expression in these discs due to the very large clone size induced by $Ras^{V12}$. The volcano plot was constructed in R, with the "EnhancedVolcano" package (https://github.com/kevinblighe/EnhancedVolcano). GO-term analysis was performed with the *Drosophila* database at https://geneontology.org/.

## DamID and bioinformatics

Genomic DNA was extracted from wandering third instar larval eye discs of *ato-GAL4 > UAS-Dam* control, *ato-GAL4 > UAS-Gl-Dam, ato-GAL4 > UAS-PntP1-Dam, elav-GAL4 > UAS-Dam* control, *elav-GAL4 > UAS-Gl-Dam*, and *elav-GAL4 > UAS-Pnt-Dam* flies, using 100 animals per replicate and three replicates per genotype. Eye discs were dissected in PBS and stored at −20 °C in PBS for up to 3 months. To extract genomic DNA, 100 pairs of eye discs were homogenized in 200 µl ice-cold 0.1 M Tris pH 9/0.1 M EDTA/1%SDS (solution A). Another 300 µl of ice-cold solution A was added and the samples were incubated for 25 min at 70 °C. 70 µl of 8 M potassium acetate was added and samples were incubated on ice for 30 min. The samples were centrifuged at 4 °C at 19,800×*g* for 15 min and the supernatants were extracted with an equal volume of phenol-chloroform-isoamyl alcohol mixture 49.5:49.5:1 (Sigma-Aldrich 77618), followed by a chloroform extraction. DNA was precipitated from the supernatant with 0.5 volume of isopropanol, incubated at room temperature for 5–10 min, and centrifuged for 15 min at 4 °C. The pellet was washed with 1 ml of 70% ethanol, air-dried, and

resuspended in 50 µl water overnight. RNAse treatment was performed before Dpn1 digestion as described[20]. A small proportion (~5 µl) of the genomic DNA for each replicate was run on a 0.8% agarose gel to check integrity, and samples with DNA smears were discarded. Further processing was done as in ref. 20. MyTaq™ HS DNA Polymerase (Meridian Bioscience, cat. no. Bio-21112) was used to amplify the adaptor-bound DNA. Cycling conditions used were 72 °C for 10 min, 95 °C for 30 s, 65 °C for 5 min, and 72 °C for 15 min; Repeat 3×: 95 °C for 30 s, 65 °C for 1 min, 72 °C for 10 min; Repeat 17×: 95 °C for 30 s, 65 °C for 1 min, 72 °C for 2 min; 72 °C for 5 min, 4 °C hold. Sonicated DNA was run on an Agilent Tape Station for QC, yielding an average segment size of 200-300 bp.

For library preparation, samples were first cleaned with Ampure bead cleanup. 1.5 volumes (150 µl) of XP cleanup was added and 50 µl was eluted and used as library input. Library preparation was performed via NEBNext® Ultra™ II with four PCR cycles. Sequencing was performed via SP100 cycle flow cell on a NOVA Seq 6000. All of the reads from the sequencing experiment were mapped to the reference genome (dm6) using Bowtie2 (v2.2.4)[68] and duplicate reads were removed using Picard tools (v.1.126) (http://broadinstitute.github.io/picard/). Low-quality mapped reads (MQ < 20) were removed from the analysis. The read per million (RPM) normalized BigWig files were generated using BEDTools (v.2.17.0) and the bedGraphToBigWig tool (v.4). Peak calling was performed using MACS (v1.4.2)[69] and peak count tables were created using BEDTools. Differential peak analysis was performed using DESeq2 and ChIPseeker (v1.8.0)[70]. R package was used for peak annotation. To compare the level of similarity among the samples and their replicates, we used two methods: principal-component analysis and Euclidean distance-based sample clustering. Heatmaps were generated using Matlab (2022b). UpSet diagrams were plotted in R studio using the "UpSetR" package. Significant peaks were called with |log2-fold change | >1 and FDR < 0.1 compared to the Dam control. Pie charts and bar graphs were made using Prism 10. MEME-Suite STREME was used to discover motifs enriched in these peaks with bed files as inputs. DamID peak visualizations for specific genes were done using IGV: Integrative Genomics Viewer and ATAC-Seq visualizations were done with the UCSC genome browser (http://genome.ucsc.edu/s/cbravo/Bravo_et_al_EyeAntennalDisc)[41]. A heatmap of the ATAC-Seq data was generated firstly by converting the normalized wig files downloaded from the UCSC table browser to bigwig and then bedgraph files using UCSC programs (http://hgdownload.soe.ucsc.edu/admin/exe/), then mapped to the DamID peaks with BEDTools (v.2.17.0).

## Statistics and reproducibility

No statistical method was used to predetermine the sample size. Three biological replicates were performed for RNA-Seq and DamID experiments, and 6–13 independent biological samples were used for immunostaining experiments. No data were excluded from the analyses. Samples were not randomized, as the experiments were based on genotype. Bioinformatics analysis was done by individuals unfamiliar with the expected results for each genotype. Phenotypic analysis was not carried out blind, as the differences between genotypes were large enough to prevent effective blinding. Significance was calculated as indicated in the figure legends.

## Reporting summary

Further information on research design is available in the Nature Portfolio Reporting Summary linked to this article.

## Data availability

All the raw data for scRNA-seq, bulk RNA-Seq, and DamID-Seq have been archived online with Gene Expression Omnibus (GEO) with the accession number GSE256221. Source data are provided with this paper.

## Code availability

No custom code was created for this paper. However, we have deposited the code we used to analyze our data in Github[71] (https://doi.org/10.5281/zenodo.12770071).

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

## Acknowledgements

We thank Ethan Bier, Andrea Brand, Iswar Hariharan, Christian Klämbt, the Bloomington *Drosophila* Stock Center, the Developmental Studies Hybridoma Bank, and the Vienna *Drosophila* Resource Center for fly stocks and reagents. The information available on FlyBase was invaluable for this study. We thank Gael Westby, Peter Meyn, and Adriana Heguy in NYU Langone's Genome Technology Center (RRID: SCR_017929) for coordinating DamID-Seq and RNA-Seq experimental planning, sample submission, processing, and sequencing. We are grateful to Alex Donovan, Anna Malkowska, Paola Angulo Salgado, Markus Schober, Hyung Don Ryoo, Jain Wu, and Maria Bustillo for advice on experimental planning and data processing. We thank Sophia He, Ariel Hairston, Dhaval Gandhi, and Genie Jang for technical assistance, and Yan Deng for confocal microscope training and maintenance. The manuscript was improved by the critical comments of Maria Bustillo and Neha Ghosh. This work was supported by NIH grants R21EY024826 and R21EY031442 to J.E.T., a grant from the Retina Research Foundation to G.M., and Swiss National Science Foundation grant 310030_219348 to S.G.S.

## Author contributions

Investigation: H.W., K.K.B.R., K.Y., C.A.M., A.T., P.C., A.J., C.F., J.E.T.; Data analysis: H.W., K.K.B.R., A.K.-J.; Supervision and funding acquisition: S.G.S., G.M., J.E.T.; Writing: original draft: H.W.; Writing: review and editing: K.K.B.R., K.Y., C.A.M., A.T., C.F., S.G.S., J.E.T.; Project management: J.E.T.

## Competing interests

C.A.M. is a shareholder of 10X Genomics. G.M. is the co-owner of Genetivision Corporation. The other authors declare no competing interests.
