## [Peer Review File · Nature Communications]

Synergistic activation by Glass and Pointed promotes neuronal identity in the *Drosophila* eye discReviewers' Comments:

Reviewer #1:

Remarks to the Author:

Overall, this is a solid study on neuronal fate in the *Drosophila* eye disc, which includes both scRNA-seq analysis and rich functional experiments. scRNA-seq study is an important component in the manuscript and my major comments are about the study design and the analysis of scRNA-seq data.

1. Details on the biosamples of scRNA-seq seem to be missing: In the section “10x Genomics single-cell RNA-Seq”, “30 male white prepupal gl60j eye discs were dissected and dissociated into single cells as described 20.”. However, wild type (wt) was included in Figure 1. It seems that the description for wt is missing in the methods section.
2. Batch effect and the overall experimental design. It is well known that scRNA-seq experiments are subjected to batch effects. After reading the “Materials and Methods” section, the overall experimental design for scRNA-seq seems unclear to me. “30 male white prepupal gl60j eye discs were dissected and dissociated into single cells as described 20.”: are the cells from the 30 eye discs pooled together and one single-cell experiment for all the pooled cells or one single-cell experiment for each eye disc? How is batch effect adjusted? It seems that these details are missing in the “Material and Methods” section.
3. In Figure 1B, the authors compared the proportions of different cell subpopulations between wt and gl. It seems that the result is obtained from pooling the cells across a number of individual flies. The proportions of cell subpopulations may vary across individual flies. Is the trend for the proportions of different cell subpopulations between wt and gl consistent if we compare the distribution of the proportions for individual flies between wt and gl (i.e. comparing the boxplots of the proportions of each cell type across the wt flies and gl flies)?
4. For reproducibility and data sharing, the raw and processed scRNA-seq datasets, and the major source code for analyzing scRNA-seq data should be accessible to the reviewers and to the public after the manuscript is published. Reviewer tokens in GEO should be given for reviewers to check that the datasets are publicly available.

Reviewer #2:

Remarks to the Author:

The manuscript by Wang and colleagues describes the synergistic role that the Glass and Pointed transcription factors play in specifying cell fates in the developing *Drosophila* eye. The authors use a combination of single cell RNA-seq, DamID, clonal analysis, and classical genetics to show that these two transcription factors specify fates by working at multiple layers of the eye GRN. Overall, it is a really interesting paper because it nice documents how intrinsic and extrinsic factors cooperate at the transcriptional level to regulate gene expression and cell fate specification. The paper is well written, the experiments are well done, the figures are clear and of high quality, and the conclusions are well supported by the data. I think that the manuscript is appropriate for Nature Communications. I would just ask the authors to address a few points listed below.

1. Glass expression starts relatively early in the differentiation process yet its effects appear to be mostly on late born photoreceptors and on the later steps of differentiation. Could the authors comment a bit more on what role they think Glass is playing in the early photoreceptor cells and during early stages of development.
2. The paper focuses almost exclusively on photoreceptors yet Glass is required in cone and pigment cells as well. Since the Glass/Pointed combination makes sense for the photoreceptors do the authors predict that Glass would also synergize with another factor in the other cell types. Maybe a mention of this in the discussion might be helpful.
3. What do glass/pointed double mutants look like?
4. When Glass and Pointed activate neuronal development in the wing disc, are the cells general neurons or are individual subtypes specified. Is Glass/Pointed a specific signal or is it a general signal to become a neuron or photoreceptor?

Reviewer #3:

Remarks to the Author:

This manuscript is a nice refinement in our understanding how general signal transduction such as that mediated by the EGF receptor is able to induce cell-type specific gene expression programming across various tissues. Almost 25 years ago, a trio of Cell papers by the Michelson, Carthew and Banerjee groups described shared mechanisms by which EGFR signaling induced gene programming in the Drosophila eye and muscle. Synergistic interactions at target gene enhancers between lineage restricted TFs and EGFR-responsive TFs such as Pnt accounted for an AND-gate mechanism of gene regulation. Later it was shown by the Banerjee, Carthew and Pollack groups that Glass (Gl) interacts with EGFR-responsive TFs like Pnt in a similar synergistic way to regulate those same target genes. In this manuscript, the authors broaden the search for target genes in the Drosophila eye that exhibit similar nonlinear dependence on Gl and EGFR-Pnt. Using scRNA-seq, RNA-seq, ATAC-seq as tools, they find 100s of new genes and expand the generality of the previous work to something that is more like a “general principle” rather than a “proof of principle”. Thus, it merits publication in this journal.

The comments I have are mostly minor and listed below.

My only major suggestion is to leverage the scRNA seq data to better provide a perspective on Gl mechanisms in the developing R cells that could potentially complement or even enhance the perspective on Gl they derive from the bulk RNA-seq and DamID experiments performed in the manuscript. In their scRNA seq paper from a year ago, the Mardon group analyzed the WT eye scRNA-seq data using linear dimension reduction (PCA) on subsets of scRNA data (cells) that occupied streams or clusters in UMAP representing particular R cells undergoing development. For example, see the streams in Fig 1A,C,D in this manuscript. From that PCA analysis, the authors

discovered genes whose expression dynamics showed greatest variation along the principal component 1 for each cell type. Genes with such significant loadings are ones whose expression dynamics are most aligned with the trajectory of R cell differentiation. They identified many genes using such an approach.

In the current manuscript, they perform scRNA seq on *gl* mutant eyes, and the UMAP embeddings show diminished or even abortive development of the various R cell types. The data is more sparse and therefore noisy, but perhaps they could repeat the PCA analysis on the *gl* data, binning cells according to the various R cell classes and performing PCA on each class. It would then be a matter of looking at which genes had significant loadings along PC1 and which did not. How many and which genes vary expression in both WT and *gl* mutant R cells. Which genes do not vary expression in *gl* mutant cells but do so in WT cells? How do those genes relate to the ones identified in their complementary bulk-seq and DamID experiments? Given the way in which the scRNA seq data display cell type GRNs so beautifully, this is a simple but possibly information-rich task to perform.

My other comments are minor.

1. There are gaps in the discussion of the relevant literature related to Glass and EGFR signaling via Pnt. Hayashi and Carthew (2008) showed that Glass and Pnt synergistically and directly act on the Prospero (Pros) enhancer to regulate its expression in the eye - R7 and cone cells. This work followed up studies from 2000 by the Banerjee, Carthew and Michelson labs showing that EGFR-Pnt signals integrate with lineage restricted transcription factors to trigger specific gene expression and cell fates with temporal and spatial specificity. All of the above papers are quite relevant to the current study and should be cited and described in the Introduction.

2. Related to this, they did not compare wt and *gl* mutant scRNA seq profiles for Pros expression in Figure 1, in spite of analyzing many eye genes that way. This is surprising since Pros is a validated direct target of Gl in R7 and cone cells.

3. Line 146-147 is conjecture. Although Lz is downstream of Gl, they can also act in parallel on some target genes. Please delete the sentence.

4. Figure 2 is a lovely experiment. Great with a simple design and analysis. Later, they test whether the result they observe in Figure 2 (synergy between Gl and Ras) is due to MAPK phosphorylation of Gl itself. They try to do so by mutating the putative phosphorylation site in Gl to alanine, blocking phosphorylation, if it exists. But instead of repeating the analysis as they had done in Figure 2, they looked at a potpourri of other gene targets not characterized in Figure 2. This is very perplexing as it was technically easy to do. If the authors did not repeat the experimental design as shown in Figure 2 with the Gl mutant, they should. Perhaps they did and the results were confounding. That should be reported in the paper. Alternatively, they could expand Figure 2 analysis by also showing us wt Gl and its effects on Lz, sallimus, chaoptin, etc in the wing clones. There needs to be alignment between WT and mutated Gl analysis for us to make any call as to whether Gl phosphorylation is not involved.

5. A related point, phosphomimetic mutations frequently fail to mimic phospho-proteins. A negative result with mutants like this are inconclusive. This caveat should be discussed in the manuscript.
6. The experiment outlined in Figure 3 is very nice, showing a sufficiency for Gl and Ras/Pnt in regulating the genome in non-eye tissues. Its design and execution are first rate.
7. Figure 4 describes physical mapping of Pnt and Gl binding to the genome of eye cells. A puzzling observation is that of the ~5,000 sites scored positive for Gl expressed using either the atonal or elav drivers, only ~1,000 sites are bound by Gl under both conditions, even though atonal and elav expression occurs in many cells one after the other over time. Can the authors speculate as to why this was observed? It is not consistent with other developmental systems such as stem cells where TF binding has been monitored over time.
8. I am a little puzzled as to why the authors did not compare their DamID datasets of “positive” genes with those genes identified as responsive in their Figure 3 experiment probing sufficiency. They only compare the DamID data to loss-of-function RNA seq data. Could the former not also be performed? It might provide more insights into the process. For example, of the many genes upregulated by Ras and Gl in a Pnt dependent manner, how many had DamID occupancy nearby?
9. The scrape scrt experiments show the mutants have a weak effect on R7 identity as defined by its axonal targeting. They claim the mutants had no effect on expression of Pros in R7 cells. This is a little surprising since Pros is required for proper R7 axon targeting (Kauffmann and Carthew 1995 Development). Normally Pros is expressed at a basal level in the young R7 cell and then gets upregulated to higher levels due to Sevenless signaling. This higher level is needed for R7 differentiation, including targeting. Looking at the Figure 6, I am not convinced that Pros is upregulated in the mutants - it is difficult to know unless one compares the signal to the basal Pros signal in neighboring cone cells, which the authors do not show. Can the authors show this more clearly, explain if it is upregulated in the mutants, and if it is, discuss why there is a discrepancy between Pros not affected and axon targeting.

We thank the reviewers for their thoughtful comments on our manuscript. We have responded as follows:

Reviewer #1:

Overall, this is a solid study on neuronal fate in the Drosophila eye disc, which includes both scRNA-seq analysis and rich functional experiments. scRNA-seq study is an important component in the manuscript and my major comments are about the study design and the analysis of scRNA-seq data.

1. Details on the biosamples of scRNA-seq seem to be missing: In the section “10x Genomics single-cell RNA-Seq”, “30 male white prepupal gl^{60j} eye discs were dissected and dissociated into single cells as described²⁰.” However, wild type (wt) was included in Figure 1. It seems that the description for wt is missing in the methods section.

The wild-type data used for comparison to gl^{60j} mutants were not generated in this study, but reproduced from a previously published paper (Bollepogu Raja, K. K. *et al.* A single cell genomics atlas of the *Drosophila* larval eye reveals distinct photoreceptor developmental timelines. *Nat Commun* **14**, 7205 (2023)). We now refer to this paper for a description of the methods for collection of the wild-type sample (p. 18). Both wild-type and gl^{60j} eye discs were dissociated using the same conditions and protocols.

2. Batch effect and the overall experimental design. It is well known that scRNA-seq experiments are subjected to batch effects. After reading the “Materials and Methods” section, the overall experimental design for scRNA-seq seems unclear to me. “30 male white prepupal gl^{60j} eye discs were dissected and dissociated into single cells as described²⁰.”: are the cells from the 30 eye discs pooled together and one single-cell experiment for all the pooled cells or one single-cell experiment for each eye disc? How is batch effect adjusted? It seems that these details are missing in the “Material and Methods” section.

We apologize if this was unclear. We pooled 30 gl^{60j} eye discs, dissociated them into single cells and used them for one scRNA-Seq experiment, and we have now specified this in the text (p. 18). Batch effects in scRNA-Seq experiments are relevant only when more than one experiment is performed. We scaled and normalized the data from single cells to account for cell-to-cell variation. Wild-type and gl mutant data sets were integrated using Seurat integration, which corrects for batch effects (Stuart, T. *et al.* Comprehensive integration of single-cell data. *Cell* **177**, 1888-1902 e1821 (2019)). Our manuscript now includes these details in the Methods section (p. 19).

3. In Figure 1B, the authors compared the proportions of different cell subpopulations between *wt* and *gl*. It seems that the result is obtained from pooling the cells across a number of individual flies. The proportions of cell subpopulations may vary across individual flies. Is the trend for the proportions of different cell subpopulations between *wt* and *gl* consistent if we compare the distribution of the proportions for individual flies between *wt* and *gl* (i.e. comparing the boxplots of the proportions of each cell type across the *wt* flies and *gl* flies)?

As noted in our response to point 2 above, our *gl* RNA-Seq data were obtained by pooling cells from about 15 individual flies in a single cell RNA-seq experiment. It is impossible to obtain enough cells for a single cell sequencing experiment using only two eye discs (from one fly) with 10x technologies. Even using the pooled population, an accurate quantification of *gl* mutant photoreceptor subtypes is difficult, as we show that the expression of marker genes for later-born photoreceptor types is severely affected.

4. For reproducibility and data sharing, the raw and processed scRNA-seq datasets, and the major source code for analyzing scRNA-seq data should be accessible to the reviewers and to the public after the manuscript is published. Reviewer tokens in GEO should be given for reviewers to check that the datasets are publicly available.”

We absolutely agree with the reviewer. The NCBI GEO accession number for the data is GSE256221, and the reviewer token, which we had provided to the editor with our previous submission, is gbkxoesadzgvnev. We had also sent the editor the code we used to analyze our data, and offered to provide the R files containing the actual data if we were given a link to upload these very large files. This code, which is based on previous studies, is also now available on github (https://github.com/hw1804/JTreis_Gl_Ras_project).

Reviewer #2:

The manuscript by Wang and colleagues describes the synergistic role that the Glass and Pointed transcription factors play in specifying cell fates in the developing Drosophila eye. The authors use a combination of single cell RNA-seq, DamID, clonal analysis, and classical genetics to show that these two transcription factors specify fates by working at multiple layers of the eye GRN. Overall, it is a really interesting paper because it nice documents how intrinsic and extrinsic factors cooperate at the transcriptional level to regulate gene expression and cell fate specification. The paper is well written, the experiments are well done, the figures are clear and of high quality, and the conclusions are well supported by the data. I think that the manuscript is appropriate for Nature Communications. I would just ask the authors to address a few points listed

below.

1. Glass expression starts relatively early in the differentiation process yet its effects appear to be mostly on late born photoreceptors and on the later steps of differentiation. Could the authors comment a bit more on what role they think Glass is playing in the early photoreceptor cells and during early stages of development.

We found some changes in gene expression in *gl* mutants in early photoreceptor cells and during early stages of development, but the more striking differences were seen in the late-born photoreceptors and in later stages of differentiation in all photoreceptors. It is possible that some of the effects on late-born photoreceptors are non-autonomous results of changes in the ability of early-born photoreceptors to induce the differentiation of later-born cells. For instance, expression of *rhomboid* and *roughoid* is reduced in R2 and R5 in *gl* mutants (Supplementary Table 1); these genes encode proteases which are required to process the EGFR ligand Spitz that induces later photoreceptors. We now mention this possibility (p. 6).

2. The paper focuses almost exclusively on photoreceptors yet Glass is required in cone and pigment cells as well. Since the Glass/Pointed combination makes sense for the photoreceptors do the authors predict that Glass would also synergize with another factor in the other cell types. Maybe a mention of this in the discussion might be helpful.

Indeed, our previous work identified some transcription factors that could synergize with Gl to ectopically activate cone cell genes (*Pax2*) or pigment cell genes (*Escargot*) in the wing disc. In this paper, we find that the synergy between Gl and Pnt is primarily on neuronal genes. We now mention this in the discussion (p. 13). We are actively working on understanding how Gl activates cone and pigment cell programs in distinct cell types, but capturing the process of pigment cell specification will require analysis at a later developmental stage.

3. What do glass/pointed double mutants look like?

We have now generated *gl^{60j} pnt¹⁸⁸* recombinants and made double mutant clones in third instar larval eye discs. In Supplementary Fig. 3d-m, we show that they resemble *pnt* single mutant clones in that only a single Elav-positive, Seven-up-negative cell, likely R8, differentiates in each cluster. As *gl* is primarily required in photoreceptors other than R8, it is difficult to observe its effects in the absence of *pnt*, which is absolutely required for the recruitment of those photoreceptors.

4. When *Glass* and *Pointed* activate neuronal development in the wing disc, are the cells general neurons or are individual subtypes specified. Is *Glass/Pointed* a specific signal or is it a general signal to become a neuron or photoreceptor?

The genes that are activated by *Gl* and *Pnt* in the wing disc include genes widely expressed in neurons, such as *Synaptotagmin 4* and *futsch*, genes specific for photoreceptors, such as *Pph13*, *transient receptor potential* and *Carcinine transporter*, and genes expressed in specific photoreceptor subtypes, such as *lozenge*, *sevenless* and *Lim3*. We have added a column in Supplementary Table 3 to indicate which eye disc cell types normally express the synergistically induced genes, based on our wild-type scRNA-Seq data. It is possible that the induced neurons are heterogeneous, or that they have a confused identity. We now show in Supplementary Fig. 3a-c that only a subset of the induced *Elav*-positive cells express *Seven-up*, a marker for R1, R3, R4 and R6, supporting the first possibility.

Reviewer #3:

This manuscript is a nice refinement in our understanding how general signal transduction such as that mediated by the EGF receptor is able to induce cell-type specific gene expression programming across various tissues. Almost 25 years ago, a trio of Cell papers by the Michelson, Carthew and Banerjee groups described shared mechanisms by which EGFR signaling induced gene programming in the Drosophila eye and muscle. Synergistic interactions at target gene enhancers between lineage restricted TFs and EGFR-responsive TFs such as Pnt accounted for an AND-gate mechanism of gene regulation. Later it was shown by the Banerjee, Carthew and Pollack groups that Glass (Gl) interacts with EGFR-responsive TFs like Pnt in a similar synergistic way to regulate those same target genes. In this manuscript, the authors broaden the search for target genes in the Drosophila eye that exhibit similar nonlinear dependence on Gl and EGFR-Pnt. Using scRNA-seq, RNA-seq, ATAC-seq as tools, they find 100s of new genes and expand the generality of the previous work to something that is more like a “general principle” rather than a “proof of principle”. Thus, it merits publication in this journal.

The comments I have are mostly minor and listed below.

My only major suggestion is to leverage the scRNA seq data to better provide a perspective on Gl mechanisms in the developing R cells that could potentially complement or even enhance the perspective on Gl they derive from the bulk RNA-seq and DamID experiments performed in the manuscript. In their scRNA seq paper from a year ago, the Mardon group analyzed the WT eye scRNA-seq data using linear

dimension reduction (PCA) on subsets of scRNA data (cells) that occupied streams or clusters in UMAP representing particular R cells undergoing development. For example, see the streams in Fig 1A,C,D in this manuscript. From that PCA analysis, the authors discovered genes whose expression dynamics showed greatest variation along the principal component 1 for each cell type. Genes with such significant loadings are ones whose expression dynamics are most aligned with the trajectory of R cell differentiation. They identified many genes using such an approach.

*In the current manuscript, they perform scRNA seq on *gl* mutant eyes, and the UMAP embeddings show diminished or even abortive development of the various R cell types. The data is more sparse and therefore noisy, but perhaps they could repeat the PCA analysis on the *gl* data, binning cells according to the various R cell classes and performing PCA on each class. It would then be a matter of looking at which genes had significant loadings along PC1 and which did not. How many and which genes vary expression in both WT and *gl* mutant R cells. Which genes do not vary expression in *gl* mutant cells but do so in WT cells? How do those genes relate to the ones identified in their complementary bulk-seq and DamID experiments? Given the way in which the scRNA seq data display cell type GRNs so beautifully, this is a simple but possibly information-rich task to perform.*

We have now repeated a similar PCA analysis on the *gl* mutant scRNA-Seq data and compared it to wild-type, and the results are shown in Supplementary Table 2. There are too few cells in the R1/R6 and R7 clusters in *gl* mutants for us to meaningfully separate them, making PCA analysis less useful for these cells, but looking at the genes with significant loadings along PC1 did help us to identify differences between wild type and *gl* in the differentiation of other photoreceptor cell types. We also show in Supplementary Table 2 the intersections of the PC1 gene lists for each cell type with genes identified in bulk RNA-Seq gain or loss of function experiments and in our Dam-ID experiments.

My other comments are minor.

1. There are gaps in the discussion of the relevant literature related to Glass and EGFR signaling via Pnt. Hayashi and Carthew (2008) showed that Glass and Pnt synergistically and directly act on the Prospero (Pros) enhancer to regulate its expression in the eye - R7 and cone cells. This work followed up studies from 2000 by the Banerjee, Carthew and Michelson labs showing that EGFR-Pnt signals integrate with lineage restricted transcription factors to trigger specific gene expression and cell fates with temporal and spatial specificity. All of the above papers are quite relevant to the current study and should be cited and described in the Introduction.

We apologize for this omission. These papers are indeed very relevant to our study, and we would have cited them in the original submission if not for the constraints imposed by the limits on reference numbers allowed by Nature Communications. We agree with the reviewer that they are important, and we have now found alternative cuts so that we can cite at least the papers relevant to eye development in the Introduction (p. 4).

2. Related to this, they did not compare wt and gl mutant scRNA seq profiles for Pros expression in Figure 1, in spite of analyzing many eye genes that way. This is surprising since Pros is a validated direct target of Gl in R7 and cone cells.

We apologize again and have now included *pros* panels in Supplementary Fig. 1f, g (rather than in Fig. 1, in which we focus on photoreceptor clusters and do not include cone cells in the plots).

3. Line 146-147 is conjecture. Although Lz is downstream of Gl, they can also act in parallel on some target genes. Please delete the sentence.

We have deleted the sentence as requested.

4. Figure 2 is a lovely experiment. Great with a simple design and analysis. Later, they test whether the result they observe in Figure 2 (synergy between Gl and Ras) is due to MAPK phosphorylation of Gl itself. They try to do so by mutating the putative phosphorylation site in Gl to alanine, blocking phosphorylation, if it exists. But instead of repeating the analysis as they had done in Figure 2, they looked at a potpourri of other gene targets not characterized in Figure 2. This is very perplexing as it was technically easy to do. If the authors did not repeat the experimental design as shown in Figure 2 with the Gl mutant, they should. Perhaps they did and the results were confounding. That should be reported in the paper. Alternatively, they could expand Figure 2 analysis by also showing us wt Gl and its effects on Lz, sallimus, chaoptin, etc in the wing clones. There needs to be alignment between WT and mutated Gl analysis for us to make any call as to whether Gl phosphorylation is not involved.

We apologize for not explaining the phosphorylation site mutant analysis more clearly. Because the phosphomimetic mutation was not sufficient to induce genes such as *elav* and *futsch* that are induced by the combination of Gl and Ras^{V12}, we had wanted to include genes that are induced by Gl alone, such as *chp* and *s/s*, to show that the mutant proteins were active. We did include a wild-type Gl construct with the same V5 tag (Gl^{PFSP}) to show that it could induce *chp* and *s/s* but not *lz*, and had hoped that readers would appreciate the inclusion of additional target genes. Since this rationale was not obvious to the reviewer, we have now added panels showing Elav and Futsch

staining of wing discs with clones expressing GI^{PFSP} , GI^{PFAP} , GI^{PFDP} or $GI^{PFAP} + Ras^{V12}$ as Supplementary Fig. 2t-ae.

5. A related point, phosphomimetic mutations frequently fail to mimic phospho-proteins. A negative result with mutants like this are inconclusive. This caveat should be discussed in the manuscript.

We have added this caveat (p. 7). However, we think that the ability of Ras^{V12} to synergize with GI^{PFAP} is a stronger argument that the synergy is not based on MAPK phosphorylation of this site.

6. The experiment outlined in Figure 3 is very nice, showing a sufficiency for GI and Ras/Pnt in regulating the genome in non-eye tissues. Its design and execution are first rate.

We are grateful for the reviewer's appreciation of our work.

7. Figure 4 describes physical mapping of Pnt and GI binding to the genome of eye cells. A puzzling observation is that of the ~5,000 sites scored positive for GI expressed using either the atonal or elav drivers, only ~1,000 sites are bound by GI under both conditions, even though atonal and elav expression occurs in many cells one after the other over time. Can the authors speculate as to why this was observed? It is not consistent with other developmental systems such as stem cells where TF binding has been monitored over time.

We agree that the cells that express *elav* in the eye disc have previously expressed *atonal*, and we also thought that there would be more overlap between the binding sites at these stages, although we expected to capture genes with the *ato-GAL4* driver that might no longer be bound by GI or Pnt in differentiated *elav*-expressing photoreceptors. *ato* is also transiently expressed in cells that will become cone or pigment cells and will not express *elav*, so the threshold for detecting significant binding may be different in the *ato* and *elav* subsets. It is also possible that there is some demethylation or dilution of the methylation signal by cell division after the stage of *ato* expression. As we do not have a definitive explanation, we have not included these speculations in the manuscript.

8. I am a little puzzled as to why the authors did not compare their DamID datasets of "positive" genes with those genes identified as responsive in their Figure 3 experiment probing sufficiency. They only compare the DamID data to loss-of-function RNA seq data. Could the former not also be performed? It might provide more insights into the

process. For example, of the many genes upregulated by Ras and Gl in a Pnt dependent manner, how many had DamID occupancy nearby?

We did include the DamID data for genes that were synergistically induced in the wing disc in the heat map shown in Fig. 5c. But we have now added an UpSet plot showing overlap between the DamID data and wing disc misexpression data as Fig. 4f.

9. The scrape scrt experiments show the mutants have a weak effect on R7 identity as defined by its axonal targeting. They claim the mutants had no effect on expression of Pros in R7 cells. This is a little surprising since Pros is required for proper R7 axon targeting (Kauffmann and Carthew 1995 Development). Normally Pros is expressed at a basal level in the young R7 cell and then gets upregulated to higher levels due to Sevenless signaling. This higher level is needed for R7 differentiation, including targeting. Looking at the Figure 6, I am not convinced that Pros is upregulated in the mutants - it is difficult to know unless one compares the signal to the basal Pros signal in neighboring cone cells, which the authors do not show. Can the authors show this more clearly, explain if it is upregulated in the mutants, and if it is, discuss why there is a discrepancy between Pros not affected and axon targeting.

We have not been able to detect any Pros staining in cone cells in the 48 h APF pupal retinas we show in Fig. 6, even though we see strong staining in R7 cells and mechanosensory bristle precursors. We do observe cone cell expression in the larval eye disc with the same antibody, so we assume that it is lost by mid-pupal stages. We have added the single Pros channels for *scrt* and *scrt scrape* clones as Fig. 6p and Fig. 6t, to show that there is no difference in the level of Pros expression in R7 between the mutant clones and neighboring wild-type tissue. We do not think that our data are inconsistent with published work; although Pros indeed regulates genes that are required for normal R7 axon targeting, Scrt and Scrape may regulate the same genes independently of Pros and/or regulate a different set of genes that are also necessary for R7 targeting.

Reviewers' Comments:

Reviewer #1:

Remarks to the Author:

All my comments are reasonably addressed.

Reviewer #2:

Remarks to the Author:

The authors have done a nice job addressing my concerns. I am supportive of the paper's publication in Nature Communications.

Reviewer #3:

Remarks to the Author:

The authors have successfully addressed all of my concerns and comments. It is ready for publication.